# RADA-dependent branch migration has a predominant role in plant mitochondria and its defect leads to mtDNA instability and cell cycle arrest

**Nicolas Chevigny**[1], **Frédérique Weber-Lotfi**[1], **Anaïs Le Blevenec**[1], **Cédric Nadiras**[1], **Arnaud Fertet**[1], **Marc Bichara**[2], **Mathieu Erhardt**[1], **André Dietrich**[1], **Cécile Raynaud**[3,4], **José M. Gualberto**[1]*

**1** Institut de Biologie Moléculaire des Plantes, CNRS, Université de Strasbourg, Strasbourg, France,
**2** Biotechnologie et Signalisation Cellulaire, CNRS, Université de Strasbourg, Illkirch-Graffenstaden, France,
**3** Université Paris-Saclay, CNRS, INRAE, Univ Evry, Institute of Plant Sciences Paris-Saclay (IPS2), Orsay, France, **4** Université de Paris, CNRS, INRAE, Institute of Plant Sciences Paris-Saclay (IPS2), Orsay, France

* jose.gualberto@ibmp-cnrs.unistra.fr

**Data Availability Statement:** All relevant data are within the manuscript and its Supporting Information files.The pipeline for analysis of the

## Abstract

Mitochondria of flowering plants have large genomes whose structure and segregation are modulated by recombination activities. The post-synaptic late steps of mitochondrial DNA (mtDNA) recombination are still poorly characterized. Here we show that RADA, a plant ortholog of bacterial RadA/Sms, is an organellar protein that drives the major branch-migration pathway of plant mitochondria. While RadA/Sms is dispensable in bacteria, RADA-deficient Arabidopsis plants are severely impacted in their development and fertility, correlating with increased mtDNA recombination across intermediate-size repeats and accumulation of recombination-generated mitochondrial subgenomes. The *radA* mutation is epistatic to *recG1* that affects the additional branch migration activity. In contrast, the double mutation *radA recA3* is lethal, underlining the importance of an alternative RECA3-dependent pathway. The physical interaction of RADA with RECA2 but not with RECA3 further indicated that RADA is required for the processing of recombination intermediates in the RECA2-depedent recombination pathway of plant mitochondria. Although RADA is dually targeted to mitochondria and chloroplasts we found little to no effects of the *radA* mutation on the stability of the plastidial genome. Finally, we found that the deficient maintenance of the mtDNA in *radA* apparently triggers a retrograde signal that activates nuclear genes repressing cell cycle progression.

## Author summary

In flowering plants, the mitochondrial genome is very large and dynamic, and its stability influences plant fitness and development. Rearrangements by recombination drive its very rapid evolution and can lead to valuable agronomic traits such as cytoplasmic sterility, used by breeders for the production of hybrid seeds. Here we describe RADA, a DNA

sequence data is described in https://github.com/ARNTET/Plant_organellar_DNA_recombination.

**Funding:** This work was supported by the LABEX MitoCross (ANR-11-LABX-0057_MITOCROSS), by IdEx Unistra (ANR-10-IDEX-0002) and EUR IMCBio (ANR-17-EURE-0023), under the framework of the French Investments for the Future Program. The funders had no role in study design, data collection and analysis, decision to publish, or preparation of the manuscript.

**Competing interests:** The authors have declared that no competing interests exist.

helicase essential for the stability of the mitochondrial DNA in Arabidopsis. We demonstrate that RADA has branch migrating activity, accelerating the processing of recombination intermediates. *radA* mutants are severely affected in development and fertility. They display mitochondrial genome instability that results in uncoordinated replication of subgenomes created by recombination. Furthermore, we found that an important component of the growth defects of *radA* mutants is apparently a cellular response triggered by the sensing of damages to the mitochondrial genome, resulting in the activation of genes that suppress the progression of the cell cycle. Our results underline the importance of better understanding the plant mitochondrial recombination pathways and their cross-talk with nuclear gene expression.

## Introduction

The mitochondrial genomes (mitogenome or mtDNA) of vascular plants are large and complex, mostly consisting of non-coding sequences assembled in a heterogeneous population of subgenomic molecules [1–3]. Their complexity derives from homologous recombination (HR) activities mobilizing the repeated sequences that are abundant in the mitogenomes of most higher plants. Large repeated sequences (>500 bp) are involved in frequent and reversible HR, while intermediate-size repeats (IRs) (50–500 bp) or microhomologies can promote infrequent HR or illegitimate recombination, respectively. Both the HR across IRs and the illegitimate recombination involving microhomologies are mainly asymmetric, with accumulation of only one of the possible alternative conformations, resulting from the break-induced replication (BIR) pathway [4–7]. Recombination involving IRs or microhomologies contributes to the heteroplasmic state of mtDNA, by creating sub-stoichiometric alternative configurations (mitotypes) that co-exist with the main genome and constitute a sink of genomic variability for the rapid evolution of plant mtDNA organization [8]. These mitogenome variants may express chimeric ORFs that can be deleterious for mitochondrial function, like in the case of cytoplasmic male sterility (CMS) [9,10]. HR is also the main pathway for the repair of double strand breaks (DSBs) and the copy-correction of mutations, thus contributing to the very slow evolution of plant mtDNA coding sequences [11,12].

Many of the factors involved in plant mtDNA recombination pathways are derived from prokaryotic homologs inherited from the symbiotic ancestors of mitochondria and chloroplasts [6,13]. However, the organellar pathways can significantly depart from those of their bacterial counterparts, and involve additional factors with partially redundant functions. For example, plant organellar HR relies on the abundant RecA-like RECA2 recombinase (about 450 copies/mitochondrion [14]) that is targeted to both organelles and whose mutation is lethal at the seedling developmental stage [15,16]. Organellar HR also involves the plastidial enzyme RECA1 and the mitochondrial enzyme RECA3. RECA3 could not be detected in Arabidopsis cultured cells [14], but its function is important for mtDNA maintenance, since its loss causes mtDNA instability that worsens over generations [15,16]. Double mutants of *RECA3* and other genes encoding mitochondrial recombination factors such as *MSH1* and *RECG1* are also more affected in development than the individual homozygote mutants [16,17]. This apparently reflects specialized functions and RECA3-dependent alternative recombination pathways, maybe because *RECA3* is expressed in specific tissues such as pollen and ovules [15]. It was also found that tobacco *RECA3* is cell cycle regulated [18].

In bacteria, HR is initiated by the loading of RecA on ssDNA, forming a nucleofilament that then seeks for homologies in the genome by probing multiple heterologous sequences [19,

20]. When a homologous sequence is identified, RecA-mediated ATP hydrolysis stabilizes the invading DNA, forming the synaptic complex, also known as displacement loop or D-loop. An important post-synaptic step is branch migration, which involves helicases that extend the homologous region on both sides of the D-loop, allowing the recruitment of the fourth DNA strand to form a Holliday Junction [21–23]. Three partially redundant pathways of branch migration have been described in bacteria, involving RuvAB, RecG and RadA [24,25]. RadA (also known as "Sms" for "sensitivity to methylmethane sulfonate" and not to be confused with Archea RadA that is unrelated) has long been known as a factor influencing repair by recombination [24,26], but its biochemical activities were only recently characterized [23,27]. It is an ATP-dependent ssDNA helicase composed of three functional domains: an N-terminal C4 zinc-finger, a RecA-like ATPase domain and a Lon protease-like domain [25]. In contrast to RecG and RuvAB, RadA interacts with RecA and can function in the context of the RecA nucleofilament. Bacterial *radA* or *recG* single mutants are only mildly affected in DNA repair, but the *radA recG* double mutant is severely impaired in its survival under genotoxic conditions [25]. The deficiency in multiple branch migration pathways is more deleterious to the cell than the absence of recombination, because of the accumulation of toxic unprocessed intermediates [24,25], further highlighting the crucial role of branch migration in HR.

In Arabidopsis mitochondria, RECA-dependent recombination is essential, because *recA2* mutants are seedling-lethal, and *recA2 recA3* double mutants could not be obtained [15]. Deficiency in branch-migration would thus be expected to be highly deleterious to mtDNA stability and plant viability. Plant genomes do not encode any homolog of the RuvAB branch migration complex, but code for an organellar-targeted RecG homolog (RECG1) [17,28]. However, *recG1* mutants are only mildly affected in development, suggesting that additional pathways for the maturation of recombination intermediates exist.

Here we describe a plant homolog of eubacterial RadA/Sms, potentially involved in the late steps of organellar HR pathways. We show that Arabidopsis RADA is targeted to both mitochondria and chloroplasts and that like the bacterial RadA, it is able to process recombination intermediates *in vitro*. However, contrarily to bacteria where *radA* deficiency has little impact on cell growth and survival, plant *radA* mutants are severely affected in their development, because of mtDNA instability, indicating that RADA has a more prominent role in plant mitochondrial recombination than it has in bacteria. Furthermore, we found that mtDNA instability caused by RADA deficiency triggers a retrograde response that activates genes involved in the suppression of cell cycle progression, partially explaining the growth defect of *radA* plants.

## Results

### RADA proteins are conserved in plants and structurally similar to bacterial RadA

The Arabidopsis (*Arabidopsis thaliana*) genome was screened for orthologs of RadA and the At5g50340 gene was identified as coding for a protein remarkably similar to bacterial RadA/ Sms (45% similarity), and therefore named *RADA*. Phylogenetic analysis showed that *RADA* is present in all groups of the green lineage, including land plants, green and red algae, as well as in brown algae, diatoms and in several organisms of the Stramenopile group that are not photosynthetic, such as the water mold *Phytophtora infestans* (Fig 1A). However, no RADA ortholog was found in animals or in yeast. While it is probable that plants inherited *RADA* from the prokaryote ancestor of mitochondria or chloroplasts, the high conservation of the sequences did not allow us to infer whether the ancestor was a proteobacterial or a cyanobacterial gene. Sequence alignments (Figs 1B and S1) showed that plant RADA all share the important functional motifs described for bacterial RadA/Sms [23,27]. A three-dimensional model of

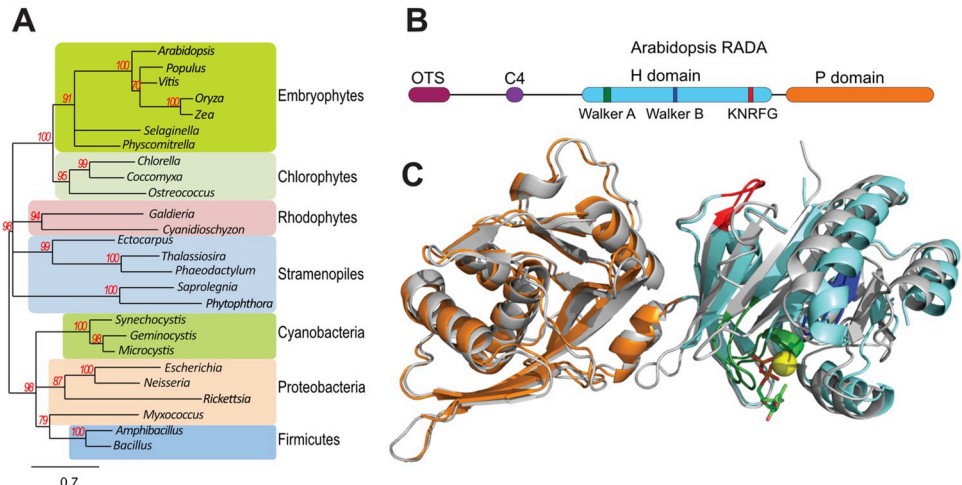

**Fig 1. RADA is an ortholog of bacterial RadA that is present in all plant lineages.** (**A**) Phenogram tree showing that genes coding for RadA-like proteins are found in all bacteria, in land plants, green, brown and red algae, diatoms and other organisms of the Stramenopile group. Bootstrap support values are indicated, and branches with values below 70% were collapsed. The scale bar indicates nucleotide substitutions per site. (**B**) The modular structure of plant RADA is similar to the one from bacteria, with a N-terminal zinc-finger (C4), a helicase domain and a Lon-protease-like domain (H and P domains, respectively). Plant precursor proteins have an N-terminal extension containing an organellar targeting sequence (OTS). (**C**) Model of Arabidopsis RADA superposed on the known structure of *S. pneumoniae* RadA (in gray) (Marie et al., 2017 [27]). The color code of relevant domains is as in (B). A bound ADP (stick representation) and the Mg$^{2+}$ ion (yellow ball) are shown.

Arabidopsis RADA was built (Fig 1C), based on the known structure of *Streptococcus pneumoniae RadA* [27]. The modeling further confirmed the very high conservation in structure of plant RADA as compared to protobacteria RadA. The RadA structure comprises an N-terminal C4 zinc-finger, required for the interaction with RecA [29], and two main domains: a RecA-like ATPase domain and a Lon protease-like domain. The RecA-like ATPase domain comprises the Walker A and B motifs, and a RadA-specific KNRFG sequence (Fig 1B). In bacteria, the Walker A and KNRFG motifs are indispensable for the branch migration function of the protein, and are also involved in the DNA-binding and helicase activities of RadA [23,27]. Walker A and KNRFG mutants are dominant negative, interfering with the function of the wild-type protein. The structural similarity between RadA and RecA suggests functional similarities, but while RecA specialized in the recognition of homologous sequences and strand invasion, RadA rather evolved for driving branch migration. Finally, the C-terminal P-domain of RadA is similar to the Lon protease-like domain of RecA, but the residues involved in protease activity are not conserved and no protease activity could be detected [30]. This domain is actually involved in DNA binding and it works as a scaffold for the protein architecture, promoting its oligomerization as hexameric rings [27,30].

## Arabidopsis RADA is targeted to both mitochondria and chloroplasts

Plant RADA sequences have non-conserved N-terminal extensions predicted to be organellar targeting peptides. We confirmed this by expression of N-terminal fusions to GFP in transgenic Arabidopsis plants. Both the N-terminal sequence of RADA as well as the full protein could target GFP to both chloroplasts and mitochondria, as shown by colocalization with the autofluorescence of chlorophyll and with MitoTracker (Fig 2A and 2B). In chloroplasts, the protein localized mainly in discrete speckles that seem to be nucleoids, according to colocalization with the nucleoid marker PEND:dsRED (Fig 2A and 2C). It is likely that in mitochondria, RADA also localizes in nucleoids. Thus, RADA is a dually targeted organellar protein, like

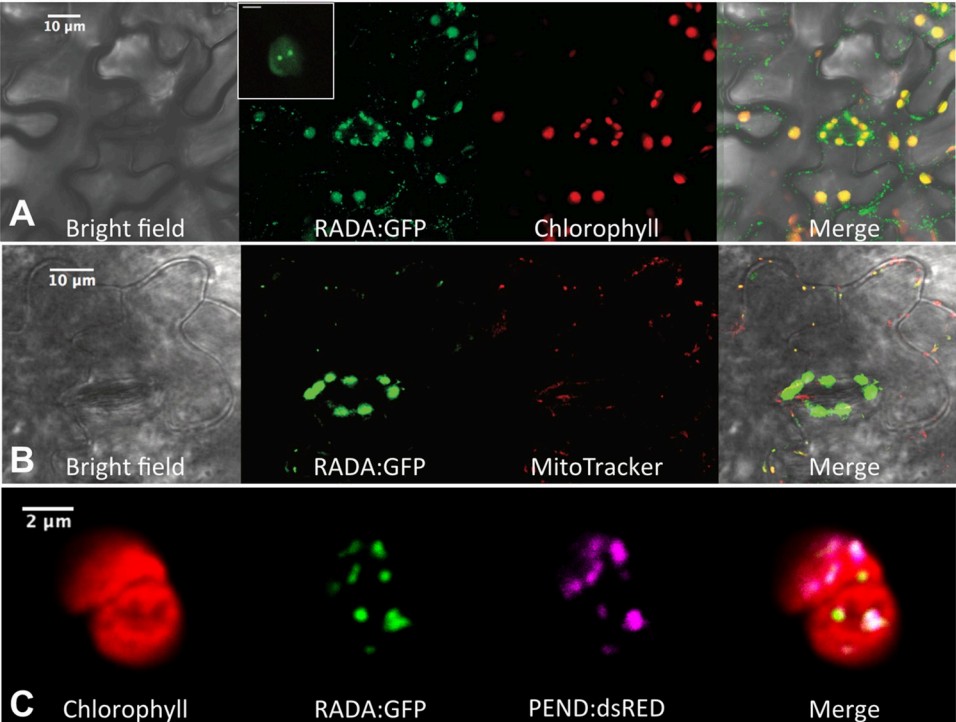

**Fig 2. Arabidopsis RADA is targeted to both chloroplasts and mitochondria.** (**A** and **B**) RADA:GFP constitutively expressed in Arabidopsis transgenic plants is double targeted to both chloroplasts (colocalization with autofluorescence of chlorophyll) and into mitochondria (characteristic cytosolic speckles and colocalization with red fluorescence of MitoTracker). The image corresponds to epidermal mesophyll cells from the abaxial side of rosette leaves. In chloroplasts RADA:GFP is predominant in dots (insert in **A**) that could be nucleoids according to colocalization with plastidial nucleoid marker PEND:dsRED (**C**).

several other factors involved in the maintenance of organellar genomes [6]. Among those factors, MutS-like MSH1 was described as having a spatially regulated localization to plastids within the epidermis and vascular parenchyma [31]. We also found RADA:GFP mainly accumulated in the epidermis and vascular tissue of the Arabidopsis rosette leaves (S2A and S2B Fig). However, we observed the same spatial distribution with the fusion comprising only the N-terminus of RADA and GFP (S2C Fig). Because this N-terminal signaling peptide is not conserved among plant RADA sequences, the differential spatial localization is apparently not linked to any RADA-specific functions. A previous report described that a rice RADA ortholog is targeted to the nucleus [32]. This was inferred from immunodetection with an antibody raised against the recombinant protein. However, our observations of RADA:GFP did not give any hint that Arabidopsis RADA is also targeted to the nucleus (Figs 2 and S2).

Expression of a promoter:GUS fusion construct revealed higher GUS expression in young shoots, in sepals and in the stigma (S3A Fig). These results are in line with expression data available at Genevestigator (https://genevestigator.com), indicating preferential expression of *RADA* in hypocotyls and in the shoot apex (S3B Fig).

## Plant RADA is an ssDNA-binding protein that stimulates branch migration in strand-exchange reactions

An Arabidopsis RADA recombinant protein was expressed in bacteria, fused to an N-terminal His-tag. A Walker A-deficient mutant protein (K201A) was also prepared. The mutation of

the equivalent position in *Escherichia coli* RadA abolished DNA-dependent ATPase activity and generated a dominant negative *radA* allele [23,25]. According to the structure of the protein bound to ADP, the mutation abolishes ATPase activity but it does not affect ATP or ADP binding [27]. Both WT RADA and K201A could be expressed and purified as soluble proteins. By gel filtration they resolved as two peaks of high molecular weight (S4A and S4B Fig), indicating different degrees of oligomerization. Dynamic light scattering showed that the smaller size oligomer was constituted by a monodispersed particle of about 340 kDa, consistent with a hexameric RADA complex (S4C Fig). EMSA experiments showed that both peak fractions could bind to an ssDNA oligonucleotide (S4D Fig). The purified proteins were tested in EMSA experiments, using constant probe concentrations and increasing concentrations of recombinant protein. Different DNA structures were tested as substrates, including ssDNA, dsDNA, fork-like structures and double-stranded molecules containing 5' or 3' ssDNA overhangs. Arabidopsis RADA could bind to all structures containing ssDNA regions. It could also bind dsDNA, but with much less affinity (Fig 3A). In our experimental conditions we found that high molecular weight complexes could be formed, which could be resolved in 4.5% polyacrylamide gels and at low voltage. These higher molecular weight complexes were promoted by the presence of ATP or ADP and could correspond to the polymerization of RADA on ssDNA, forming nucleofilaments (Fig 3B). We did not see significant differences in binding

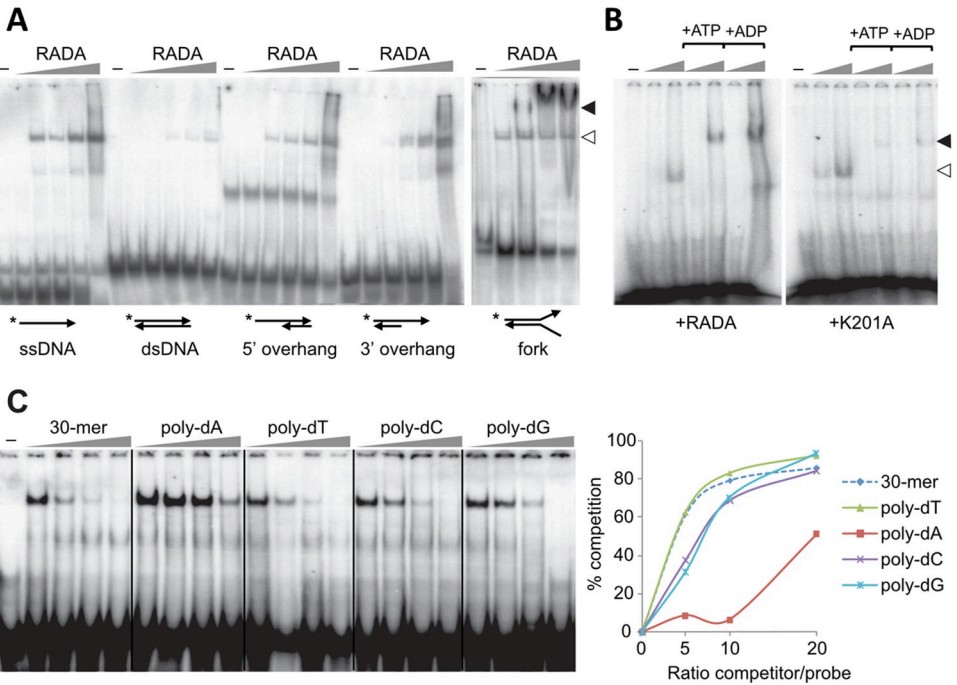

**Fig 3. RADA preferentially binds ssDNA.** (**A**) EMSA experiments showing that RADA binds to any ssDNA-containing DNA structure with higher affinity than to dsDNA. Lower and higher molecular weight complexes are indicated by white and black arrowheads respectively. A predominant band seen with the 5'-overhang probe is artefactual, already present in the absence of protein. (**B**) Analysis on low concentration gel (4.5% as compared to 8% in A) resolves the high-molecular weight RADA:ssDNA filament, which is promoted by ATP or ADP (1 mM). The K201A mutant protein binds ssDNA with an affinity comparable to the RADA WT protein. Increasing concentrations of RADA or of K201A used in (A) and (B) are indicated by the gray triangles. (**C**) Competition experiments. A 30-mer ssDNA oligonucleotide (7x[AGTC]AG) was used as probe in EMSA experiments with recombinant RADA and sequence specificity was tested by competition, with increasing concentrations of the cold homologous oligonucleotide or with 30-oligomers (poly-dA, pol-dT, poly-dC and poly-dG). Quantification of the results is shown on the right. Only poly-dA showed reduced competition for RADA binding to ssDNA.

with the K201A mutant protein, which seemed to bind ssDNA with equivalent affinity as the WT protein. The sequence specificity of RADA binding to ssDNA was tested in competition experiments. In these experiments, poly-dT, poly-dC and poly-dG could compete binding as efficiently as the homologous probe, while poly-dA was a less efficient competitor (Fig 3C). Thus, Arabidopsis RADA binds preferentially to ssDNA, with little sequence specificity.

The branch migration activity of plant RADA was tested using an *in vitro* strand-exchange reaction. In this test, bacterial RecA in the presence of ATP and of SSB initiates the invasion of dsDNA by homologous ssDNA and promotes branch migration till the final heteroduplex product is completed (Fig 4A). In the presence of plant RADA the branched intermediates were resolved faster, leading to the earlier appearance of the final nicked double-stranded circular product (Fig 4A). The faster resolution of recombination intermediates was reproducibly observed in six independent experiments (Fig 4B). We also tested whether RADA can promote branch migration independently from the presence of RecA. For that, assays were arrested by freezing at a time point (7 min) when there was already accumulation of branched intermediates, but no final heteroduplex product was visible (Fig 4C left panel). The reaction mix was deproteinated and the purified nucleic acids added to new reaction mixes, in the presence or absence of RADA. In the absence of both RecA and RADA the branched intermediates could not spontaneously evolve and remained stable (Fig 4C middle panel). However, in the presence of RADA they were converted to the final product, showing that plant RADA alone can promote branch migration (Fig 4C right panel). This departs from what was described for bacterial RadA, whose function seems to depend from its interaction with RecA. However, RADA is not a recombinase redundant with RecA, because RADA alone in the absence of RecA is not able to initiate strand-invasion (Fig 4D). Finally, in the same experimental conditions, the K201A mutant protein was unable to promote branch migration, and rather behaved like a dominant-negative, completely inhibiting the reaction (Fig 4E). The incapacity of the K201A mutant to migrate along the heteroduplexes probably blocks the activity of RecA. This result is consistent with the dominant negative effect of the equivalent mutation in *E. coli* RadA [25].

## Plant RADA can complement bacterial RadA in the repair of DNA damage

We tested the ability of plant RADA to complement *E. coli* RadA in the repair of genotoxic stress-induced DNA lesions. A *radA785(del)::kan* strain was used for complementation assays, and Arabidopsis RADA or *E. coli* RadA were expressed from the low-copy number plasmid pACYC. As previously described by others, we found the *radA* strain to be little affected by genotoxic treatments as compared to WT [24,25]. The conditions we found best to test complementation were in the presence of ciprofloxacin, an inhibitor of gyrase that induces DNA DSBs. In a spot assay, the growth of RadA-deficient cells was much reduced as compared to WT. This ciprofloxacin-triggered growth defect could be complemented by the expression of Arabidopsis RADA, as efficiently as by the expression of bacterial RadA cloned in the same expression vector (Fig 4F). Therefore, plant RADA can functionally substitute bacterial RadA in the repair of DNA damages induced by ciprofloxacin, in agreement with the similar *in vitro* activities of the bacterial and plant proteins.

## Arabidopsis plants deficient in RADA are severely affected in their development and fertility

Two Arabidopsis T-DNA lines (*radA-1* and *radA-2*) could be validated, with T-DNA insertions in exons 8 and 5 respectively and both in the Col-0 accession background (Fig 5A). Homozygous plants from both mutant lines displayed severe developmental defects such as retarded growth and distorted leaves showing chlorotic sectors (Fig 5B). The phenotype was

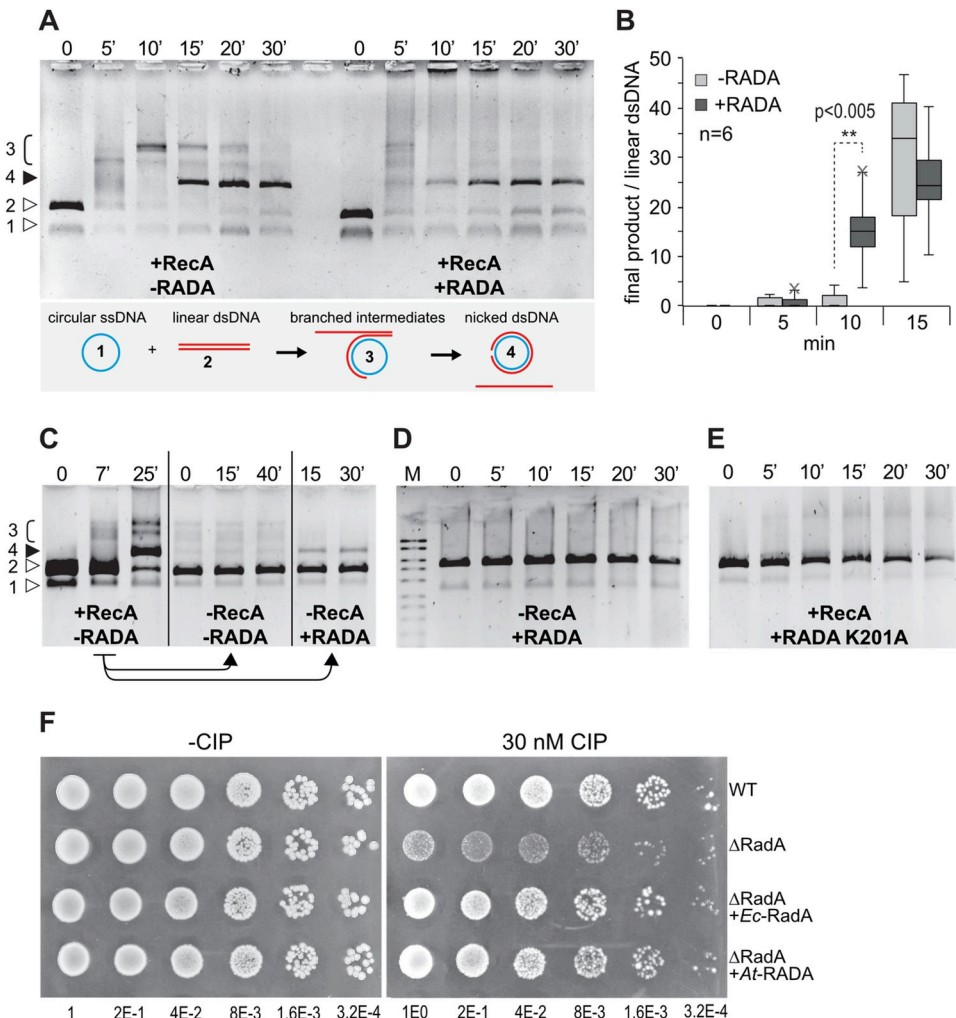

**Fig 4. Branch-migration activities of RADA.** (**A**) Recombinant RADA accelerates branch-migration of DNA heteroduplexes initiated by RecA, in an *in vitro* reaction. An explanation of the different substrates and products is shown. (**B**) Ratio of final product as compared to the initial linear dsDNA substrate in 6 independent experiments, showing that in the presence of RADA there is faster resolution of branched intermediates. (**C**) Experiment showing that RADA alone can finalize branch-migration initiated by RecA: a reaction at t = 7 min was arrested by deproteination (left panel) and the DNA purified. Without further addition of RecA or RADA proteins there is no spontaneous progression of the reaction (middle panel), but added RADA can alone resolve the recombination intermediates into the final product (right panel). (**D**) RADA alone cannot initiate strand invasion. (**E**) Mutation of the ATPase Walker domain of RADA (K201A) inhibits the reaction. (**F**) Complementation of an *E. coli radA* mutant (ΔRadA) for growth in the presence of genotoxic ciprofloxacin (CIP). Arabidopsis *RADA* (*At*-RADA) complements the mutation as efficiently as the bacterial protein (*Ec*-RadA) cloned in the same expression vector. Serial dilutions of the cell suspensions were spotted and are indicated in the x-axis.

fully penetrant, with all homozygous mutants displaying this phenotype, although some plants were more severely affected than others. To fully confirm that the observed phenotypes were because of a deficiency in RADA, *radA-1* plants were complemented with the WT *RADA* gene expressed under its own promoter. Heterozygous *radA-1* plants were transformed and homozygous *radA-1* plants that also contained WT *RADA* as a transgene were segregated in the T2 generation. They were phenotypically normal (Fig 5B, right), confirming the complementation and the linkage of the growth defects to RADA loss of function.

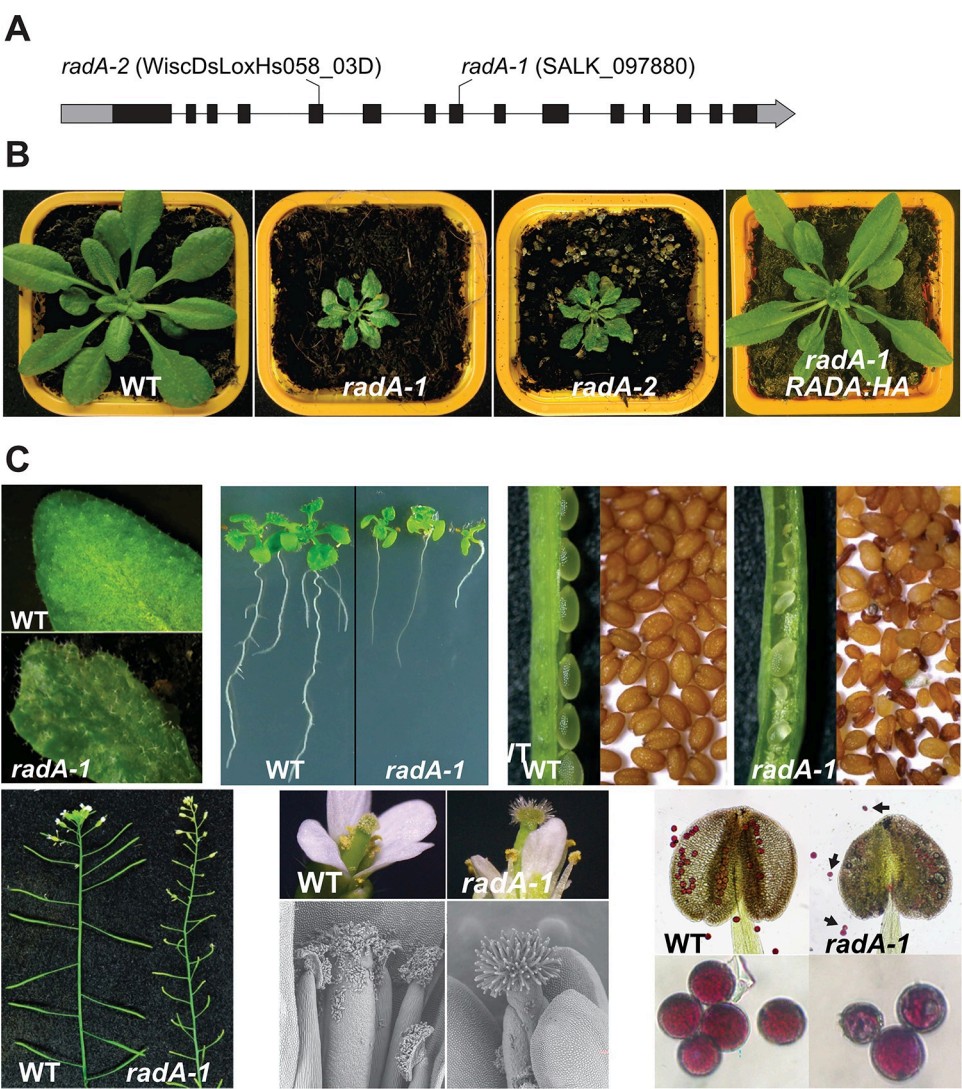

**Fig 5. Arabidopsis *radA* mutants and phenotypes.** (**A**) Schematic representation of the Arabidopsis RADA gene. Coding sequences are in black and 5'- and 3'-UTRs are in gray. The position of the T-DNA insertions in *radA* mutant lines is shown. (**B**) radA plants show severe growth retardation, with distorted leaves presenting chlorotic sectors. These phenotypes can be complemented by expression of HA-tagged RADA (RADA:HA) under control of the endogenous *RADA* promoter. (**C**) Detail of mutant phenotypes as compared to WT, showing deformed leaves, root shortening and flower stems with very small *radA* siliques that mostly contain aborted seeds. Most of the few seeds produced are non-viable. Flowers are also deformed, and visible and SEM images show that in *radA* no pollen binds to the papillae of the stigma. Alexander staining of pollen in *radA* anthers as compared to WT shows little pollen production and an abundance of small and aberrant pollen grains (indicated by arrows).

Details of *radA* plants phenotypes are shown in Fig 5C. As compared to WT the mutants have small and distorted leaves, short roots, very small siliques mostly containing aborted seeds, and most of the few seeds produced are non-viable. Flowers are also distorted, and visible and SEM images show that virtually no pollen binds to the papillae of the stigma. That could be because papillae cells are modified and unable to bind pollen, or because the stigma develops and is receptive before pollen maturation. Pollen stained positive by Alexander staining, suggesting that it is viable. That was confirmed by the successful transmission of the *radA* allele in crosses using *radA* pollen. However, pollen production was much reduced compared

to WT, and many pollen grains were of aberrant size and shape. To test whether *radA* female gametes are viable, emasculated flowers from WT or *radA* were pollinated with WT pollen and ovules were observed before and after pollination (S5 Fig). The mature unfertilized *radA* ovules looked morphologically normal. Three days after pollination virtually all WT ovules were fertilized and showed a developing embryo, but only 1/6th of the *radA* ovules had developing embryos. No elongation of the pollinated pistils was observed even at seven days after pollination, suggesting that normal pollen could not fertilize the apparently normal *radA* ovules, potentially because of a deficiency in pollen germination on the stigma of *radA* plants. Thus, the partial sterility of *radA* could be due to both male and female defects.

Because RADA is targeted to mitochondria and plastids, we observed if the organelles are phenotypically normal in *radA* mesophyll cells, by transmission electron microscopy (S6 Fig). The TEM images showed that in *radA* cells, chloroplasts are morphologically normal and indistinguishable from those of WT leaf cells. On the other hand, mitochondria looked enlarged in size and less electron dense than those from WT, suggesting that it is the mitochondrion that is predominantly affected in *radA*. Bigger mitochondria were also reported in other mutants impaired in mitochondrial gene expression or genome maintenance, such RNAi lines affected in the expression of organellar RNase P and in mutants of SHOT1 that is apparently involved in the organization of mitochondrial nucleoids [33,34].

## *radA* mutants are affected in the stability of the mitochondrial genome

Several Arabidopsis mutants affected in mtDNA recombination functions (ex: *msh1*, *osb1*, *recA2*, *recA3*, *recG1*) have mitogenome instability because of increased ectopic recombination across IRs [6,35,36]. The *radA* mutants were thus also tested for such molecular phenotype. Illumina sequence of the DNA extracted from a pool of *radA* plants revealed a coverage profile of the mtDNA displaying dramatic changes in the relative stoichiometry of mtDNA sequences, as compared to the mostly homogeneous coverage of the mtDNA in WT (Fig 6A). Several of those regions are flanked by the large repeats LR1 or LR2, and by pairs of directly oriented IRs (ex, pair of repeats A [556 bp], F [350 bp], L [249 bp] and EE [127 bp]), suggesting that the process at play is the looping out of circular subgenomes by recombination, followed by their autonomous replication (Fig 6A).

To assess if these changes in mtDNA stability correlate with the developmental phenotype, we analyzed individual plants (Fig 6B). Four seedlings grown *in vitro* were selected, according to the severity of the growth defects, and the relative copy number of the different mtDNA regions quantified by qPCR, using a set of primer pairs spaced about 5 kb apart across the genome, as described previously [17]. Changes in the relative stoichiometry of mtDNA sequences were observed in all plants, mostly overlapping with the ones identified by DNAseq (Fig 6B). However, the amplitude of the changes was higher in the severely affected plants (*radA-1#1* and *radA-2#1*) than in the mildly affected ones (*radA-1#2* and *radA-2#2*). As compared to WT, an increase in copy number of large genomic regions could be observed, as high as seven-fold. We tested by qPCR the accumulation of alternative conformations resulting from recombination across repeats F, L or EE and, as expected, in all plants there was a significant increase in several of such products *versus* WT levels (Fig 6C), with a significantly higher accumulation in the more affected plants than in the mildly affected ones. Recombination resulted in the asymmetrical accumulation of mainly one of the two possible products, which could be because of alternative mtDNA repair by the break-induced replication (BIR) pathway [6,7]. Big differences in the relative accumulation of recombination products were seen, depending on the pair of repeats analyzed. However, these values are misleading because they are the ratio to the basal levels that exist in WT plants. Thus, a 30-fold increase in

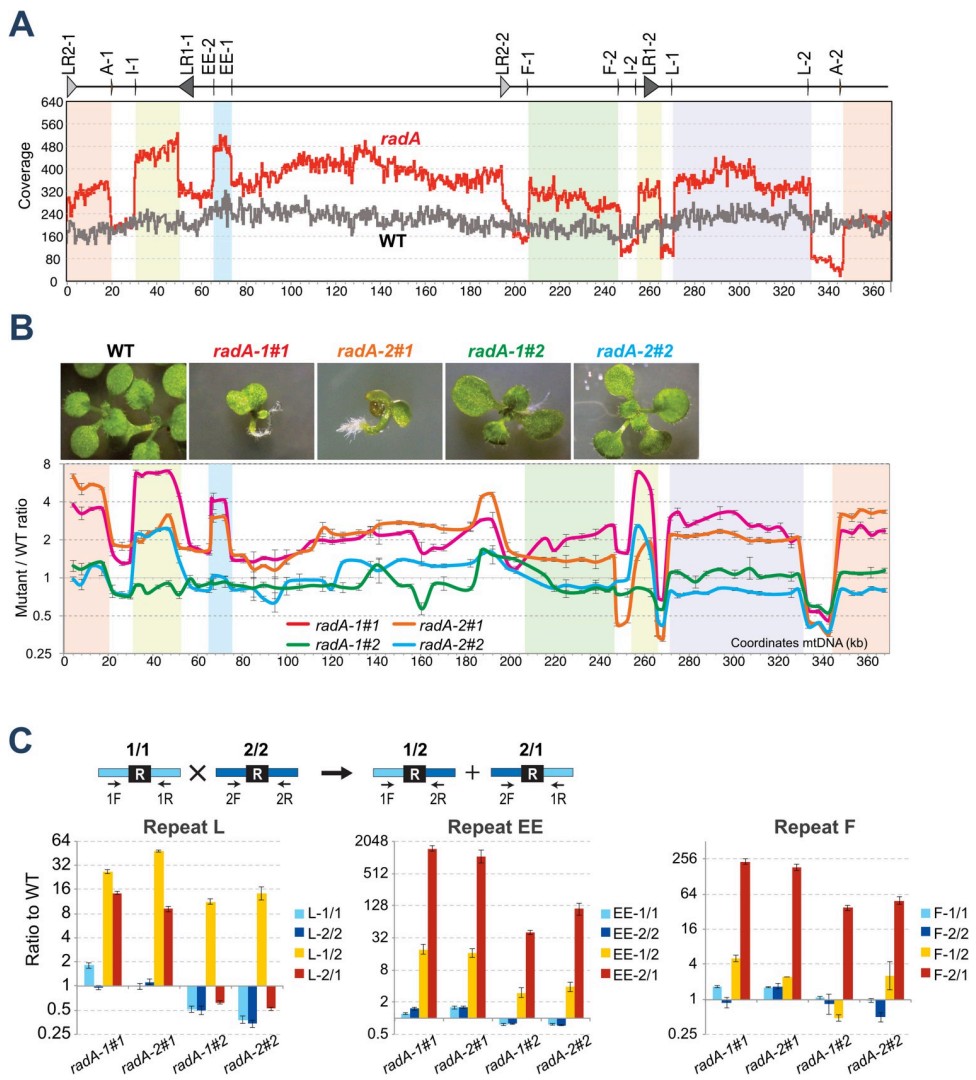

**Fig 6. Changes in mtDNA sequences stoichiometry in *radA* mutants because of increased recombination across intermediate size repeats.** (**A**) Comparison of Illumina sequence coverage of the mtDNA in *radA-1* as compared to WT. Coordinates are those of the Col-0 mtDNA sequence. The position of the mtDNA large repeats LR1 and LR2 and of relevant IRs are shown above. Regions with changed stoichiometry that are flanked by repeat pairs are shadowed. (**B**) Picture of severely affected (*radA-1*#1 and *radA-2*#1) and mildly affected (*radA-1*#2 and *radA-2*#2) seedlings and scanning of their mtDNA for changes in relative copy numbers of the different mtDNA regions. Sequences spaced 5–10 kb apart on the mtDNA were quantified by qPCR. Regions with changed stoichiometry flanked by repeat pairs are shadowed as in A. (**C**) Accumulation of recombination products across repeats L, F or EE in *radA* seedlings. The scheme above shows the qPCR relative quantification of parental sequences 1/1 and 2/2 and of the corresponding alternative arrangements 1/2 and 2/1. Results are in a log2 scale and error bars correspond to *SD* values from three technical replicates.

recombination product L-1/2 might be equivalent in absolute copy number to a 1000-fold increase in product EE-2/1, because the former is already quite abundant in WT Col-0 and easily detected by hybridization, while the latter is virtually absent in WT [17,35].

Because RADA is also targeted to chloroplasts, the plastidial DNA (cpDNA) of *radA* plants was likewise scanned for changes in sequence stoichiometry. No changes were detected between mutant and WT (S7A Fig). We also checked whether *radA* plants have increased rearrangements of the cpDNA by recombination involving microhomologies, as described for

*why1why3* and *why1why3reca1* Arabidopsis mutants and for maize *cptK1* [37–39]. For that we developed a pipeline for identification of Illumina reads corresponding to cpDNA rearrangements, using soft-clipping information (S7C Fig). The *radA* sequence libraries contained an equivalent proportion of rearrangements as compared to WT (S7B Fig), less than one event per cpDNA copy, *i.e.* too low to be responsible for the severe growth defects of *radA* plants.

The full-length *RADA* gene expressed under its own promoter can fully complement the *radA* growth defects (Fig 5B). To further explore possible effects of RADA on cpDNA maintenance, we tested hemicomplementation of the *radA* mutant with constructions allowing specific targeting of RADA either to mitochondria (AOX1:RADA) or to the chloroplasts (RBCS:RADA) (S8A Fig). These targeting sequences have been shown to be sufficient for specific and efficient organellar targeting [40]. Recently, it was reported that the RBCS targeting sequence might leak into mitochondria in conditions of overexpression [41]. However, such problem shouldn't affect our experiments, using the endogenous *RADA* promoter. The constructs were introduced in heterozygous *radA-1* plants, and homozygous mutants containing the hemicomplementation constructs were selected (hereafter referred to as *radA AOX1:RADA* and *radA RBCS:RADA* respectively, S8B Fig). None of the hemicomplementation constructs could achieve complete complementation of the *radA* growth defect phenotype. We could not compare the transgenes expression levels, as the HA-tagged transgene-deriving proteins could not be detected in total plant extracts from either leaves or inflorescences. Nevertheless, *radA AOX1:RADA* plants were less affected in growth than *radA RBCS:RADA* plants and did not present the deformed leaves phenotype observed in non-complemented *radA* plants. Analysis of the recombination involving pair of repeats L also revealed decreased accumulation of alternative product L-1/2 in *radA AOX1:RADA* as compared to *radA-1* or to *radA RBCS:RADA* plants (S8C Fig). Taken together, DNAseq and hemicomplementation results support the hypothesis that the severe growth phenotype of *radA* plants results from defects in mtDNA maintenance, and that the loss of RADA has no significant effects on the stability of the cpDNA.

## RADA is the predominant branch migration pathway of plant organelles

In bacteria, the *radA* mutation is highly synergistic with *recG* [23,24]. We have therefore tested the epistatic relationship between Arabidopsis *RADA* and *RECG1*, the Arabidopsis ortholog of bacterial RecG. In our study we used the *recG1-2* mutant that is a true knock-out of *RECG1* [17]. To compare all mutant plants at the first homozygous generation, heterozygous *recG1-2* plants were crossed with *radA-1* heterozygous plants used as pollen donor. In the segregating F2 population, we obtained WT, *recG1-2*, *radA-1* and *recG1-2 radA-1* double homozygous mutants (Fig 7A). In this cross, plants inherited the organellar genomes of accession Ws, which is the genetic background of *recG1-2*. As previously described, the *recG1-2* single mutants were phenotypically normal [17]. The *radA–1* single mutant developed the same growth defects observed in the Col-0 background. Surprisingly, the *recG1-2 radA-1* double mutants were as severely affected in growth as the *radA-1* single mutant, with no evidence for an additive effect of the two mutations (Fig 7A). A negative synergistic epistatic relation was expected if both genes were involved in alternative, partially redundant pathways, as in bacteria. Since *recG1 radA* double mutants are as severely affected in growth as *radA* plants, RADA is apparently the predominant branch migration pathway in plant organelles.

## RADA is a player in the RECA2-dependent mitochondrial pathway, but apparently not in the one that relies on RECA3

The mtDNA recombination pathways depend on two RecA-like recombinases, RECA2 that is essential and RECA3 that is dispensable [15]. It was shown that *recG1* is synergistic with *recA3*

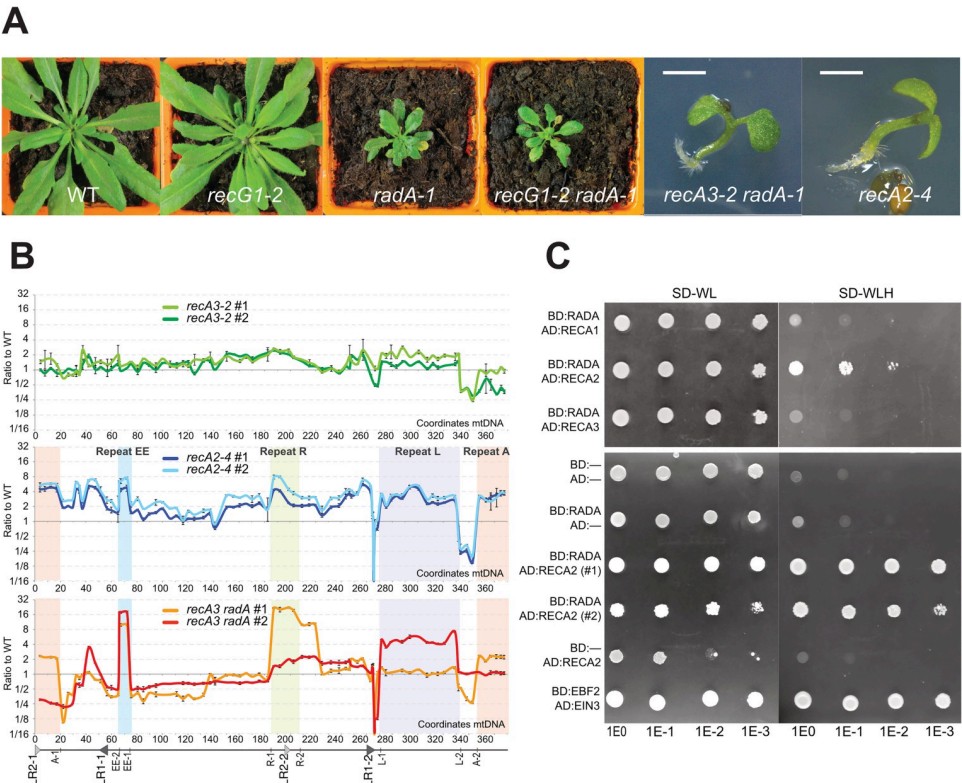

**Fig 7. Synergistic effect of *radA* with *recA3* but not with *recG1* is consistent with a role in the RECA2-dependent pathway.** (**A**) Crosses of *recG1-2* and *radA-1* (pollen donor) showed that double homozygote *recG1 radA* plants are as affected in development as simple homozygote *radA* plants. Contrarily, double homozygote *recA3 radA* do not grow roots and are lethal at the seedling stage, a phenotype similar to the one of *recA2*. The scale bar is 1 mm. (**B**) qPCR scanning of relative copy numbers of the different mtDNA regions, as described in Fig 6, in pools of *recA3*, *recA2 and recA3 radA* seedlings. Results were compared to those of WT seedling of same size, and are represented in log2 scale. (**C**) RADA interacts with RECA2, but not with RECA1 or with RECA3, on a yeast two-hybrid assay. Serial dilutions of cell suspensions were spotted. BD and AD are the binding and activating domains of GAL4, respectively. Replicate experiments with two independent yeast colonies co-transformed with both plasmids (#1 and #2) confirmed this result, and controls with empty vectors showed that there was no autoactivation by either BD:RADA or AD:RECA2 (lower panel). The interaction between the F-box protein EBF2 and EIN3 was used as positive control.

[17], suggesting that RECG1-dependent branch migration is only important in one of the alternative pathways. Therefore, we also tested the epistatic relationship between *radA* and *recA3*. Heterozygous *recA3-2* and *radA-1* (both in Col-0 background) were crossed, but no double homozygous mutants could be retrieved from F2 plants growing on soil. Seeds from sesquimutant *recA3-/- radA+/-* were germinated *in vitro* and seedlings were genotyped, revealing that *recA3 radA* double homozygous mutants can germinate, but are unable to grow roots and to expand their cotyledons (Fig 7A). This seedling lethal phenotype was similar to the one observed for *recA2* [15]. The effects of *recA2*, *recA3*, and *recA3 radA* mutations on the stability of the mtDNA were analyzed, as described for *radA* in Fig 6B. This was done on *recA3* and *recA3 radA* first-generation homozygous mutant plants segregated from the same crosses, and compared to WT seedlings of same size. The molecular analysis showed that mtDNA stability was greatly affected in *recA3 radA*. While segregating *recA3* single mutant plants showed only mild effects on the stoichiometry of mtDNA sequences, in *recA3 radA* plants there were dramatic changes, some regions being more than 20-fold reduced or increased as compared to neighboring sequences (Fig 7B). The magnitude of the changes was higher than the one

observed for highly affected *radA* plants of third homozygote generation (Fig 6B). The profiles were equivalent to the one obtained for *recA2* plants, with both *recA2* and *recA3 radA* showing dramatic reduction of the regions between repeats L-2 and A-2 and between large repeat LR1-2 and repeat L-1. Because these sequences are duplicated in the nuclear genome of Col-0 [42], most probably the reduction observed corresponds to a complete loss from the mtDNA. Concordant with this assumption, a small sequence (278 bp) unique to the mtDNA downstream LR1-2 is absent in both *recA2* and *recA3 radA*. However, these lost sequences do not contain any known essential gene. As previously found for *recA2* [15], the copy number of the cpDNA was also not significantly affected in *recA3 radA* (S9 Fig). Thus, the seedling lethality of *recA3 radA* and of *recA2* correlate with massive problems in the stoichiometric replication and segregation of the mtDNA.

The synergistic effect of the *radA* and *recA3* mutations suggests that the two factors intervene in alternative pathways, with RADA predominantly acting in the essential RECA2-dependent recombination pathways, explaining why the loss of both RADA and RECA3 phenocopies the loss of RECA2. In bacteria, the function of RadA involves direct interaction with RecA. We have therefore tested if plant RADA has affinity to any of the three organellar RECA proteins of Arabidopsis. RECA1, RECA2 and RECA3 were tested by yeast two-hybrid for interaction with RADA, either fused to the binding or activating domains of GAL4 (BD and AD respectively). We observed interaction between BD:RADA and AD:RECA2, but not with AD:RECA1 or AD:RECA3 (Fig 7C, upper panel). Replicate experiments confirmed this result, and controls with empty vectors showed that there was no autoactivation by either BD: RADA or AD:RECA2 (lower panel). Thus, the apparent interaction between RADA and RECA2 but not with RECA1 or RECA3 further supports that branch-migration by RADA is predominantly important in the mitochondrial RECA2-dependent recombination pathway, but apparently not in the RECA3-dependent one, or in plastidial recombination that relies on RECA1.

## *radA*-induced mtDNA instability has no apparent negative effects on the accumulation of mitochondrial transcripts

The instability of the mtDNA in *radA* plants correlated with the severity of the growth defect phenotypes, but in no case we did find a significant reduction in the copy number of mtDNA genes, potentially linking phenotype to a defect in gene expression. The relative abundance of the mitochondrial gene transcripts and rRNAs was quantified by RT-qPCR, but no apparent defect in mtDNA gene expression was found. Rather, for most transcripts an increased accumulation was observed, as compared to WT plants of the same size (Fig 8A), up to 8-fold in the case of the *rps4* transcript. To test whether *radA* plants are deficient in the assembly of OXPHOS complexes, these were analyzed by Blue-Native gel electrophoresis. However, no visible defects were detected, neither on Coomassie stained complexes nor by activity staining of Complex I and immunodetection of Complex III, which revealed complexes of the same size and abundance (Fig 8B).

## Instability of the mtDNA in *radA* mutants affects cell cycle progression

Scanning electron microscopy (SEM) showed that *radA* epidermal leaf cells were much enlarged as compared to cells of WT leaves (Fig 9A). The *radA* leaves also had significantly fewer stomata (384.mm$^{-2}$ in *radA-1 versus* 779.mm$^{-2}$ in WT, $\chi^2$ test = 8.5E-14). This suggested an inhibition of cell division in *radA*. To test such a possibility, nuclear DNA ploidy levels were measured by flow cytometry, in the fully developed first true leaves of 20-day-old *radA* and WT plants (Fig 9B and 9C). *radA* leaves displayed a higher proportion of 4C and 8C nuclei

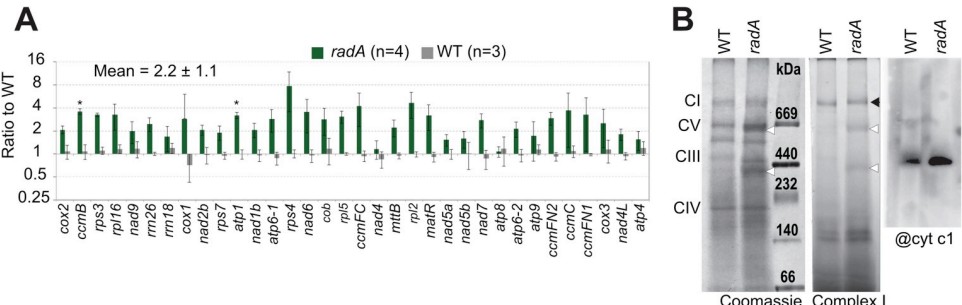

**Fig 8. Accumulation of mitochondrial transcripts in *radA*. (A)** Representative mitochondrial transcripts were quantified by RT-qPCR, from the RNA of 10-day-old seedlings grown *in vitro*, and normalized against a set of nuclear housekeeping genes. Results are on a log2 scale and are the mean from four biological replicates (two pools of *radA-1* and two pools of *radA-2* seedlings) with the corresponding *SD* error bars. The asterisks indicate significance of $p < 0.01$ by Student's t test. **(B)** Blue-Native gel analysis of purified mitochondria from WT and *radA-1* plants. 80 µg of proteins were loaded in each well, and run in parallel with size markers (High Molecular Weight Calibration Kit, GE Healthcare). The identifiable OXPHOS complexes (CI, CII, CIV and CV) are indicated. Coomassie staining, complex I activity staining and immunodetection of complex III, with a cytochrome c1-specific antibody, revealed no apparent deficiency in the assembly of OXPHOS complexes in *radA*. The white arrowheads indicate green-yellow bands that were already visible before staining and likely corresponded to contaminating plastidial complexes. The black arrowhead indicates the complex I activity band.

than WT leaves, and an important reduction in 16C nuclei, suggesting that endoreduplication is inhibited in *radA*. In all *radA* samples, an accumulation of intermediate peaks (ex. 8-16C) was observed, but not in the WT, also suggesting a blockage of cell cycle progression in S phase, or an induction of programmed cell death. To further explore whether mtDNA instability in *radA* impacts cell cycle progression, we quantified nuclear DNA replication in root tips, using the thymidine analog 5-ethynyl-2'-deoxyuridine (EdU). In *radA*, the proportion of EdU-positive nuclei was significantly reduced (Fig 9D). Counting of mitotic events in root tips also showed much reduced numbers of cells undergoing mitosis in *radA* (Fig 9E).

Cell cycle checkpoints adjust cellular proliferation to changing growth conditions, via the inhibition of the main cell cycle controllers that include cyclin-dependent kinases (CDKs) [43]. In plants, *SMR* genes encode inhibitors of CDK-cyclin complexes that are transcriptionally induced in response to a variety of stress conditions, integrating environmental and metabolic signals with cell cycle control [44–46]. Interestingly, chloroplastic defects have been shown to induce cell cycle arrest through the induction of *SMR5* and *SMR7* [37,46]. How these genes are activated remains to be fully elucidated, but the mechanism could involve the activation of the DNA damage response (DDR). This signaling cascade is controlled by the ATM and ATR kinases [47] that phosphorylate the SOG1 transcription factor, leading to the up-regulation of thousands of target genes, including *SMR5* and *SMR7* [48,49]. We have therefore tested the expression of several cell cycle markers and of several genes that are transcriptionally activated as part of the DDR. These included 14 *SMR* genes, selected cell cycle-related genes and the NAC-type transcription factors ANAC044 and ANAC085 that inhibit cell cycle progression in response to DNA damage through their activation by SOG1, but also in response to heat stress through a SOG1-independent pathway [50]. Among the 14 SMR genes tested we found strong induction of *SMR5* in *radA* (Fig 9F). But expression of *SMR7*, whose activation has been described as associated with that of *SMR5* in the case of chloroplastic defects [44,46], remained unchanged. ANAC085 was strongly activated in *radA*, more than 30-fold as compared to WT, as well as ANAC044, but to a much lower degree (Fig 9F). We also observed a low-level induction of the DDR genes *BRCA1* and *RAD51*, and of the *CYCB1;1* gene encoding a cyclin associated with G2 arrest and DSB repair [51].

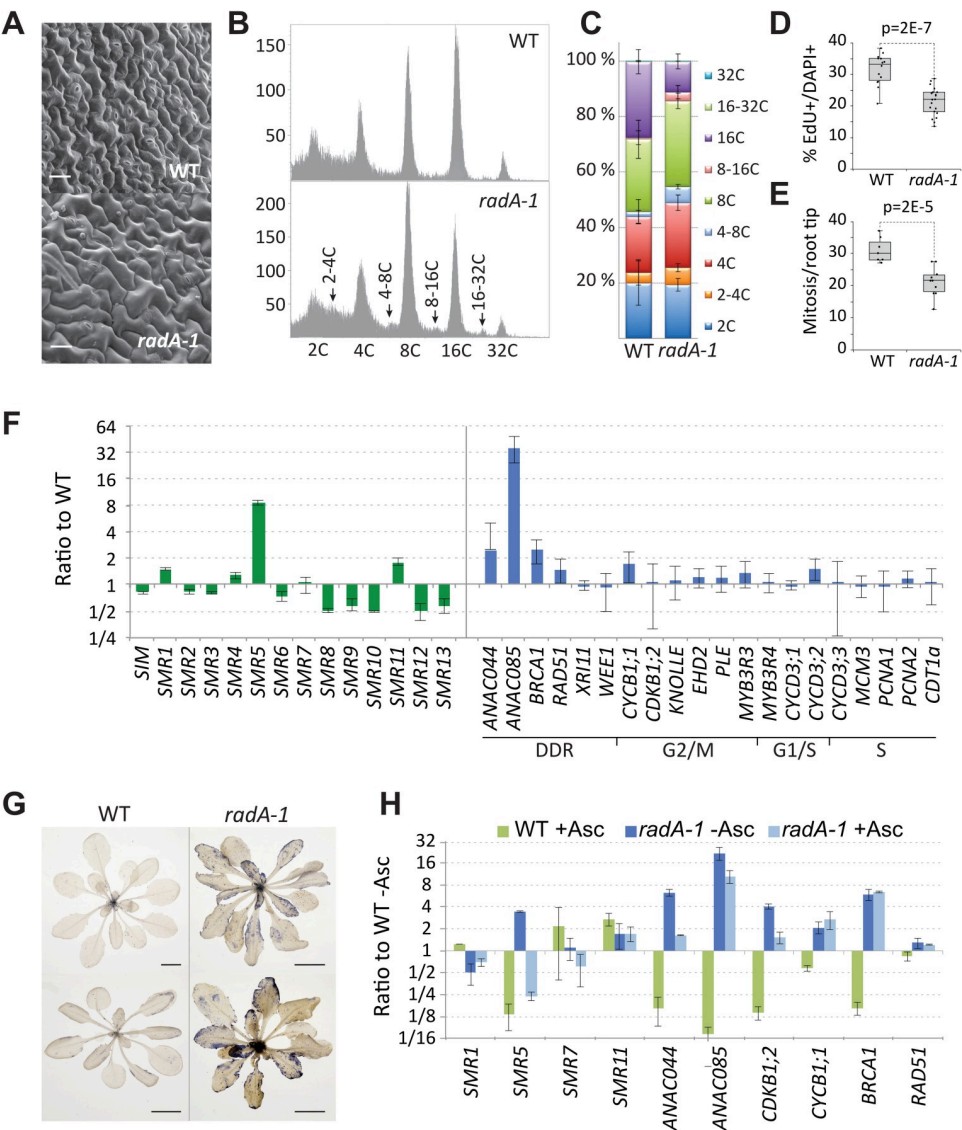

**Fig 9. Cell cycle progression is impaired in *radA* plants.** (A) Scanning electron microscopy images showing much larger epidermal cells and fewer stomata in *radA* leaves. Scale bar is 20 μm. (B) Flow-cytometry profiles of WT and *radA*. The DNA content of nuclei extracted from the first true leaves of 20-day-old plants was analyzed. (C) Ploidy distribution, showing decreased endoreduplication in the mutant, with an increased proportion of 4C and 8C nuclei and a decreased proportion of 16C nuclei. Values are the mean ± *SD* from 4 experiments for WT and from six experiments for *radA* (n>20 000 nuclei). (D) Decreased DNA synthesis in the nuclei of *radA* root tip cells, as evaluated by the ratio between EdU positive and DAPI positive cells. (E) Decreased number of cells undergoing mitosis. Significances were calculated by Student's t test. (F) RT-qPCR analysis of the expression of a set of cell cycle related genes in 10-day-old WT and *radA* seedlings, revealing significant activation of *SMR5* and *ANAC085*. The data is represented in a log2 scale and is the mean ± *SD* of three biological replicates. (G) NBT (nitro blue tetrazolium) staining for O$^{2-}$, in plants of equivalent size grown under the same conditions, showing accumulation of ROS in *radA* plants. Scale bar is 1 cm. (H) The expression of several cell cycle-related genes is suppressed by the ROS quencher ascorbic acid (1 mM), both in WT and in *radA* 10-day-old seedlings.

SMR5 is a cyclin-dependent kinase inhibitor that is induced by different conditions leading to oxidative stress [52]. ROS-dependent transcriptional activation of *SMR5* and of *SMR7* was confirmed in several ROS-inducing conditions [44]. We have therefore tested whether the induction of *SMR5* in *radA* mutants could be due to the accumulation of ROS. Whole rosettes

of WT and *radA* plants of the same size were stained with nitro blue tetrazolium (NBT) to reveal $O^{2-}$, and a much higher accumulation of ROS was indeed observed in *radA* plants (Fig 9G). To confirm the effect of ROS on the activation of cell cycle regulators, *radA* and WT plants were grown in the presence of the ROS quencher ascorbic acid, and the expression of selected genes tested. In both WT and *radA* seedlings, the expression of *SMR5*, *ANAC044*, *ANAC085* and *CDKB1;2* was significantly reduced, as compared to plants grown in the absence of ascorbate, up to 8-fold in the case of *SMR5* (Fig 9H). Thus, a component of the *radA* growth phenotype is apparently a retrograde response that activates cell cycle regulators to inhibit cell proliferation. This could be because of the release of ROS as a consequence of mtDNA instability.

## Discussion

The role of RadA (or Sms) in bacterial HR has been described only recently [23,27]. RadA is a hexameric helicase loaded by RecA on either side of the D-loop, to allow hybridization of the invading ssDNA with the recipient DNA [27]. Our results show that plant RADA apparently has similar activity in organelles, but a more essential role in genome maintenance than its bacterial counterpart. A previous report proposed that plant RADA is a nuclear protein [32], but our results only showed RADA targeted to mitochondria and chloroplasts, where it probably localizes in nucleoids. This is consistent with the presence of predicted targeting sequences in all plant RADA proteins. In rosette leaves, the GFP-tagged protein was mainly visible in the epidermis and vascular parenchyma, as described for MSH1 [31]. The plastids of these cells are smaller and have proteomes distinct from those of mesophyll chloroplasts [53]. However, GFP fusion of the N-terminus of RADA had the same spatial localization, suggesting that it simply reflects differential targeting efficiency of organellar proteins in different cell types, regardless of proteins functions.

The modeled structure of plant RADA is very close to the known structure of bacterial RadA, suggesting that the activities are also conserved. Indeed, Arabidopsis RADA could complement the survival of the bacterial *radA* mutant under genotoxic conditions, as efficiently as the *E. coli* protein when brought in *trans*. Similarly, we showed that plant RADA preferentially binds ssDNA and accelerates the *in vitro* strand-exchange reaction initiated by RecA, as described for bacterial RadA [23]. Like bacterial RadA, the Arabidopsis protein was not able to initiate strand invasion and could only promote branch-migration. This result contradicts a previous report that suggested that rice RADA could promote D-loop formation [32]. In that report, a different assay system was used, based on the invasion of supercoiled plasmid by a labeled oligonucleotide, and the efficiency seemed very low. With the Arabidopsis recombinant protein and in our test system we could not reproduce such an activity.

Nevertheless, several important differences in activity were observed between the plant and bacterial proteins. Plant RADA and the Walker A K201A mutant could bind to ssDNA-containing molecules in the absence of ADP or ATP, and both formed high molecular weight complexes in the presence of ATP or ADP, while in bacteria, ATP activates the translocation of RadA and causes its dissociation from DNA [27]. Departing also from what was described for bacterial RadA, we found that plant RADA alone can promote branch migration of recombination intermediates *in vitro*, in the absence of RecA, while the interaction with RecA was described as necessary for bacterial RadA activity.

In bacteria, the mutation of a single branch migration factor is only slightly detrimental for cell growth and DNA repair, and that is particularly true for RadA, whose loss was found deleterious only when combined with the recG or ruvAB mutations [24,25]. In plants however, loss of RADA alone severely affects plant development and fertility, and the double mutation

*radA recG1* is not synergistic. This result also contrasts with the lack of a notable developmental phenotype for the Arabidopsis *recG1* mutants [17]. It therefore seems that, in plants, RADA has a predominant and more important role than RECG1 in the processing of recombination intermediates.

In bacteria, the *radA recG* double mutation is more detrimental to cell survival than the *recA* single mutant, indicating that the accumulation of unprocessed branched intermediates is more damaging than the lack of recombination. In agreement with this hypothesis, the bacterial triple mutant *recA radA recG* is less affected in survival than *radA recG* [25]. In plants, the *recA2* mutant is lethal at the seedling stage, and the double mutant *recA2 recA3*, deficient in both mitochondrial RecA-like proteins, could not be obtained [15]. This indicates a more vital role of recombination in plant mitochondria than in bacteria. It is therefore surprising that the mutation of all known branch migration pathways is not more deleterious to plant mitochondria. It suggests that a further alternative pathway for the processing of recombination intermediates exists in plant organelles. That could be one of the functions of RECA3. Indeed, the synergistic effect of the *radA recA3* double mutation recalls the double mutants of bacterial branch migration factors [25]. Thus, it might be that RECA3 is able to process the recombination intermediates initiated by RECA2, thanks to its branch-migration activity that is intrinsic to RecA-like proteins [54]. Furthermore, RECA3 is characterized by the absence of an acidic C-terminal sequence that is found in all other RecA-like proteins, including RECA1 and RECA2 [15,16]. In bacteria, the C-terminus is a site for interaction with many other proteins that regulate RecA activity and its deletion enhances almost every one of the RecA functions [54]. Thus, RECA3 might have evolved to display enhanced branch migration activity and to be partially redundant to RADA. In the absence of RADA, processing of the HR intermediates produced by the RECA2 pathway would require activation of the alternative RECA3-dependent pathway. The loss of both RADA and RECA3 would overload the system with unresolved recombination intermediates, which would be lethal for the plant. The accumulation of toxic unresolved structures could also be the cause for the severe growth inhibition observed in later generations of *radA*, the most affected plants showing seedling lethality and root growth inhibition similar to *recA2* and *recA3 radA*. In the nucleus of eukaryotes, paralogs of RAD51, the functional homolog of bacterial RecA, have also been described as involved in the regulation of recombinase functions [55,56]. Consistent with the hypothesis that RECA3 defines an alternative minor recombination pathway, the protein could not be detected in the proteome of an Arabidopsis cell culture, while RECA2 was found as an abundant protein, about 462 copies per mitochondria [14]. RADA was quantified up to 36 proteins per mitochondria, a ratio of about 1 copy of RADA to 13 RECA2 copies, consistent with its role as a branch migration factor recruited by the RECA2 nucleofilament. The Y2H interaction of RADA with RECA2 but not with RECA3 or RECA1 further supports the assumption that RADA is mainly active in the RECA2-dependent HR pathway of plant organelles.

We found that loss of RADA elicits mtDNA recombination across IRs. This could be because of the repair of DSBs by error-prone break-induced replication (BIR), triggered by a deficiency in HR functions required for accurate replication-coupled repair [6,7]. It apparently leads to the formation of sub-genomes that replicate autonomously and uncoordinated from the remaining mtDNA, as it was observed in *recG1* for the episome resulting from recombination involving repeats EE [17]. We could not reveal any deleterious effects in the maintenance of the chloroplast genome. As discussed elsewhere, that might be because of the absence of IRs in the cpDNA of Arabidopsis [6]. Mutants of *RECG1* and *RECA2* that are also dually targeted to mitochondria and chloroplasts showed no problems in cpDNA maintenance neither.

However, although the loss of RADA can lead to massive reshuffling of the mtDNA, no region of the mtDNA containing functional genes is lost, and all mitochondrial genes that

were tested are expressed. The expression of tRNAs has not been tested, but no region comprising a tRNA gene is lost in *radA* plants. Thus, we could not correlate the developmental phenotypes with the reduced expression of a OXPHOS gene, or for a factor required for OXPHOS complex assembly. The BN gel analysis did not revealed any obvious defect in OXPHOS complexes assembly either. Rather, the severe developmental phenotypes elicited in *radA* mutants seem to partially result from a mitochondrial retrograde signal that promotes inhibition of cell cycle progression. The cell cycle is an energy demanding process that can be arrested by defects in respiration or photosynthesis [57]. Several studies have shown that chloroplastic defects lead to altered cell cycle regulation, including reduced cell proliferation and premature endoreduplication [37,46,58]. Altered progression of the cell cycle occurs in *radA*, as suggested by the increased size of epidermal cells and confirmed by determination of nuclear ploidy, analysis of EdU incorporation and mitotic activity. Our results thus show that defects in the maintenance of mtDNA integrity can also interfere with cell cycle progression in plants. Consistently, expression of *RADA* is higher in proliferating tissues, although it does not seem to be cell cycle regulated according to data generated with synchronized BY-2 cells [18]. A reduced mitotic index has also been observed in *recA3 msh1* double mutants, which develop a severe growth defect phenotype similar to *radA* [16]. Likewise, in Drosophila, mitochondrial dysfunction activates retrograde signals, including ROS, to modulate cell cycle progression [59].

In chloroplast dysfunction, accumulation of ROS is responsible for the cell cycle arrest observed in *reca1why1why3* mutants [37]. Similarly, it is possible that mitochondrial genome instability in *radA* results in sub-optimal function of the OXPHOS complexes and in ROS production, triggering cell cycle arrest and the developmental phenotypes observed. But functional deficiency of OXPHOS complexes does not *per se* activate regulators of cell cycle, since mutants of complex I display no evidence for the activation of genes such as *SMR5* or *SMR7* [60]. In chloroplast-deficient mutants such as *crl* and *reca1why1why3*, ROS accumulation is thought to activate the DDR master regulator SOG1, leading to cell cycle arrest and activation of DNA repair [37,46,58]. Indeed, DDR mutants and particularly *sog1*, are hypersensitive to genotoxins specifically targeting organelles [58], and ROS accumulation leads to SOG1 activation through its phosphorylation by the ATM kinase [44]. However, recent results showed that SOG1, ATM and ATR are not required for the activation of *SMR5* and *SMR7* in the *crl* mutant, in which chloroplast homeostasis is severely compromised [61]. This indicates that alternative pathways exist to control cell cycle progression depending on chloroplast activity. In line with this hypothesis, we found that in *radA* mutants, a number of SOG1 targets are only slightly mis-regulated. In addition, treatment with the ROS scavenger ascorbate only partially rescued the activation of genes involved in cell cycle arrest in the mutant. Interestingly, we observed that *ANAC085*, a direct SOG1 target that is directly activated by abiotic stresses, such as heat stress [50], is strongly upregulated in the *radA* mutant, and that this induction is largely independent of ROS accumulation. It is thus tempting to speculate that retrograde signaling triggered by mitochondrial genome instability also relies on alternative pathways independent of canonical DDR signaling mediated by ROS accumulation (Fig 10). Further work will be required to fully decipher how defects in organellar genomes maintenance can lead to inhibition of cell cycle progression.

## Materials and methods

### Plant materials

Arabidopsis T-DNA insertion mutant lines, all in the Col-0 background, were obtained from the Nottingham Arabidopsis Stock Centre (*radA-1*: SALK_097880, *radA-2*:

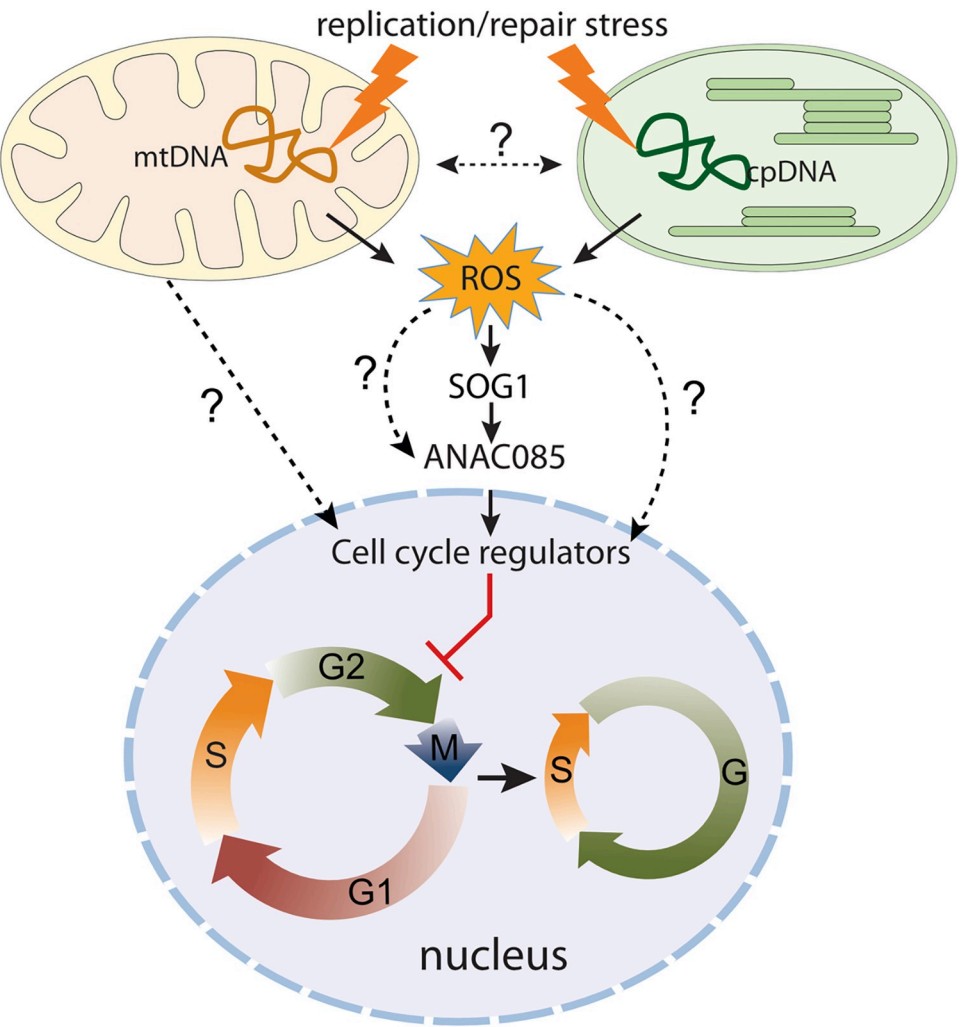

**Fig 10. Model for possible retrograde regulation by organellar DNA instability.** Possible retrograde effect of the instability of organellar genomes on the expression of cell-cycle regulators inhibiting plant growth and development. The release of ROS from mitochondria and/or chloroplast could be the signal transmitted to the nucleus, activating among others genes under control of SOG1.

WiscDsLoxHs058_03D). Seeds were stratified for 3 days at 4°C and plants were grown on soil or on half-strength MS medium (Duchefa) supplemented with 1% (w/v) sucrose, at 22°C. Plant genotypes were determined by PCR using gene and T-DNA specific primers. DNA was extracted using the cetyltrimethylammonium bromide method. RNA was extracted using TRI Reagent (Molecular Research Centre, Inc.). For mutant complementation, the WT *RADA* gene and promoter sequence was cloned in binary vector pGWB613, fused to a C-terminal 3xHA tag, and used to transform heterozygous *radA–1* plants. Expression of the transgene in the T1 transformants was monitored by western-blot with a HA-specific antibody. For hemi-complementation, i) the promoter sequence and 5' UTR of *RADA*, ii) the N-terminal sequence (49 codons) of soybean alternative oxidase 1 (AOX1; NM_001249237) or the N-terminal sequence (80 codons) of the small subunit of ribulose bisphosphate carboxylase (RBSC; At1g67090), and iii) the RADA sequence minus its first 86 codons fused to a C-terminal HA epitope and Nos terminator, were assembled together by MultiSite Gateway cloning in vector pB7m34GW,0 (https://gatewayvectors.vib.be/), giving constructs AOX1:RADA and RBCS:

RADA respectively. These constructs were used to transform heterozygous *radA-1* plants by floral dip. For promoter:GUS fusion expression analysis, the *RADA* promoter and 5'-UTR (1114 bp) was cloned in vector pMDC162 and the construction was introduced into Arabidopsis Col-0 plants.

### Intracellular localization

The cDNA sequence coding the N-terminal domain of RADA (first 174 codons) was cloned into the binary vector pBIN+, to obtain the N-RADA:GFP construct. For full protein fusion to GFP (RADA:GFP), the gene sequence and 5'-UTR (from -156 to 4172 relative to the start codon) was cloned in vector pUBC-paGFP-Dest (http://n2t.net/addgene:105107). Arabidopsis Col-0 plants were transformed by the floral dip method and leaves of selected transformants were observed on a Zeiss LSM700 confocal microscope. The fluorescence of GFP and chlorophyll was observed at 505 to 540 nm and beyond 650 nm, respectively. For mitochondrial colocalization, leaves were infiltrated with a 1/1000 dilution of MitoTraker orange (Thermo Fisher Scientific) solution. Excitation was at 555 nm and observation at 560–615 nm. Nucleoid colocalization was tested by biolistic co-transfection in *Nicotiana benthamiana* epidermal leaf cells with a PEND:dsRED construct [62].

### Isolation of mitochondria and Blue Native PAGE

Arabidopsis mitochondria were prepared from the aerial part of three-week old plants grown on soil, by differential centrifugation and step density gradients as described previously [60]. Arabidopsis mitochondrial complexes were solubilized with 1.5% dodecylmaltoside and analyzed on BN-PAGE 4–13% (w/v gels, as described [63]. Blue staining and in gel NADH dehydrogenase activity were performed as described [64].

### *In vitro* strand exchange reaction and DNA Binding Assays

Recombination assays were performed with single-strand linear ΦX174 virion DNA and double strand circular ΦX174 RFI DNA (New England Biolabs) linearized with PstI, in 20 mM Tris-acetate pH 7.4, 12.5 mM phosphocreatine, 10 U/mL creatine kinase, 3 mM ammonium glutamate, 1 mM dithiothreitol, 2% glycerol and 11 mM magnesium acetate. In our conditions, 20.1 μM (in nucleotides) linear ssDNA, 6.7 μM RecA (New England Biolabs), 2 μM RADA are incubated with buffer for 8 min at 37˚C. Then, 20.1 μM (in nucleotides) linear dsDNA is added and the whole reaction is incubated for 5 min at 37˚C. Finally, strand exchange is initiated by adding 3 mM ATP and 3.1 μM SSB (Merck). Aliquots are stopped at indicated times by addition of 12 μM EDTA and 0.8% SDS. Strand exchange products were analyzed on 0.8% agarose gels run at 4˚C and after migration visualized by ethidium bromide staining. For reactions terminated in the absence of RecA, the RecA-initiated strand exchange reaction was stopped at the indicated time and DNA was deproteinized by phenol-chloroform extraction followed by ethanol precipitation. The DNA pellet was solubilized in reaction buffer, 2 μM RADA was added and the reaction was further incubated at 37˚C for the indicated time, before quenching with 12 μM EDTA and 0.8% SDS.

For electrophoretic mobility shift assays (EMSA) the purified recombinant protein (50–500 fmol according to the experiment) was incubated with oligonucleotide probes (0.01 pmol-10 fmol) 5'-radiolabeled with $[\gamma\text{-}^{32}P]$ATP (5000 Ci/mmol; PerkinElmer Life Science). Different dsDNA structures were prepared by annealing the radiolabeled sense oligonucleotides with a twofold excess of unlabeled complementary oligonucleotide and purified on non-denaturing polyacrylamide gels. The binding reactions were performed in 20 mM Tris-HCl pH 7.5, 50 mM KCl, 5 mM MgCl$_2$, 0.5 mM EDTA, 10% glycerol, 1 mM DTT and protease inhibitors

(Complete-EDTA; Roche Molecular Biochemicals), incubated at 20°C for 20 min and run on 8 or 4.5% polyacrylamide gels in Tris-Borate-EDTA buffer at 4°C. For competition assays, labeled probe and unlabeled competitor were added simultaneously to the reaction mixture. Radiolabelled bands were revealed using a Typhoon phosphorimager (GE Healthcare Life Sciences).

## Recombinant proteins

The Arabidopsis *RADA* cDNA sequence minus first 48 codons was cloned in bacterial expression vector pET28a, fused to an N-terminal His-tag. A mutant was prepared to express a Walker A-deficient protein (K201A). The corresponding constructs pET28-RADA and pET28A-RADA[K201A] were used to express recombinant proteins in *E. coli* Rosetta 2 (DE3) pLysS (Novagen). The recombinant RADA and RADA[K201A] proteins were affinity purified in a precalibrated HisTrap FF Crude (GE Healthcare Life Sciences) column run at 0.5 mL/min, washed with 50 mM Tris-HCl pH 8.0, 300 mM NaCl, 5% glycerol, 50 mM imidazole and eluted with a 50–500 mM imidazole gradient. The recombinant protein fractions were further purified by gel filtration on Superdex S200 columns and aliquots were flash frozen in liquid nitrogen and stored at -80°C. RADA concentration was determined by spectrophotometry (extinction coefficient 42,440 $M^{-1}$ $cm^{-1}$).

## Bacterial complementation

The BW25113 *radA+* and JW4352 *radA785(del)::kan* strains were used for complementation assays. The Arabidopsis RADA cDNA (sequence coding for amino acids 137 to 627) or the *E. coli* RadA/Sms sequence were cloned between the PstI and BamHI restriction sites of the pACYCLacZ vector [15] under the control of the *lac* promoter. Both constructs were introduced in the JW4352 strain. The pACYCLacZ empty vector was introduced in the BW25113 and JW4352 strains as control. Bacteria were grown in LB supplemented with 10 μg/mL chloramphenicol till $OD_{600nm}$ = 0.4 before addition of 2.5 mM IPTG. At $OD_{600nm}$ = 1.2 bacteria were diluted $10^4$ fold and grown on LB agar plates supplemented with 15 nM ciprofloxacin, 2 mM IPTG and 10 μg/mL chloramphenicol.

## Yeast two-hybrid assays

The cDNA sequences of Arabidopsis RADA, RECA1, RECA2 and RECA3 minus the first 44, 51, 38 and 40 codons, respectively, were cloned into pGADgwT7-AD and pGBKgwT7-BD vectors containing the GAL4 activation and binding domains (AD and BD respectively). The yeast strain AH109 was co-transformed with different combinations of AD and BD vectors and transformants were selected on synthetic defined (SD)/-Leu/-Trp (SD-LW) medium for 4 days at 30°C. A single colony for each combination was inoculated in liquid SD-LW medium and grown over-night. Cultures were adjusted to an $OD_{600nm}$ of 0.5 and serial dilutions were plated on SD-LW and SD/-Leu/-Trp/-His (SD-LWH) media. As positive control we tested the known interaction between the F-box protein EBF2 and EIN3 [65].

## qPCR and RT-qPCR analysis

qPCR experiments were performed in a LightCycler480 (Roche) in a total volume of 6 μL containing 0.5 mM of each specific primer and 3 μL of SYBR Green I Master Mix (Roche Applied Science). The second derivative maximum method was used to determine Cp values and PCR efficiencies were determined using LinRegPCR software (http://LinRegPCR.nl). Three technical replicates were performed for each experiment. Results of qPCR and RT-qPCR analysis

were standardized as previously described [17]. The recently corrected Arabidopsis Col-0 mtDNA sequence was taken as reference [66]. Quantification of mtDNA and cpDNA copy numbers used a set of primer pairs located along the organellar genomes, as described previously [17,39]. Results were normalized against the UBQ10 (At4G05320) and ACT1 (At2G37620) nuclear genes. The accumulation of ectopic HR products involving IRs was quantified using primers flanking each pair of repeats, as described [15]. The COX2 (AtMG00160) and 18S rRNA (AtMG01390) mitochondrial genes and the 16S rRNA (AtCG00920) chloroplast gene were used for normalization. For RT-qPCR experiments, 5 μg of RNA were depleted from contaminating DNA by treatment with RQ1 RNase-free DNase (Promega) and reverse-transcribed with Superscript IV Reverse Transcriptase (Thermo Fisher Scientific) using random hexamers. The GAPDH (At1G13440) and ACT2 (At3G18780) transcripts were used as standards.

## Sequencing

Total leaf DNA of WT and *radA-1* plants was quantified with a QuBit Fluorometer (Life Technologies) and libraries were prepared with the Nextera Flex Library kit, according to manufacturer's recommendations (Illumina) using 100 ng of each DNA sample. Final libraries were quantified, checked on a Bioanalyzer 2100 (Agilent) and sequenced on a Illumina Miseq system (2 × 150 paired end reads).

## Flow cytometry and EdU staining

Nuclear DNA content was measured in leaves of 20-d-old seedlings, using the CyStain UV Precise P Kit (Partec) according to the manufacturer's instructions. Nuclei were released in nuclei extraction buffer (Partec) by chopping with a razor blade, stained with 4',6-diamidino-2-phenylindole (DAPI) buffer and filtered through a 30 μM Celltrics mesh (Partec). Between 20,000 and 30,000 isolated nuclei were used for each ploidy level measurement using the Attune Cytometer and the Attune Cytometer software (Life Technologies). At least four independent biological replicates were analyzed. EdU staining was as described [67]. For each root tip (n>10), the number of mitotic events was counted directly under the microscope.

## Bioinformatics analysis

Sequence alignments were constructed with T-Coffe implemented in the Macvector package. Targeting predictions were tested using the SUBA web site (http://suba.plantenergy.uwa.edu. au). For phylogenetic analysis, sequences were aligned with T-Coffe implemented in the Macvector package with default parameters (Myers-Miller dynamic algorithm, gap open penalty of -50 and gap extension penalty of -50). A consensus maximum likelihood tree was built with IQ-TREE [68], from 1000 bootstrap iterations (http://iqtree.cibiv.univie.ac.at). Graphical representation and edition of the tree were performed with TreeDyn (v198.3). The Arabidopsis RADA structure was modeled on the structure of RadA from *Streptococcus pneumoniae* (pdb: 5LKM), using Modeller (http://salilab.org/modeller/about_modeller.html). For Illumina sequence analysis of the mtDNA or cpDNA, reads were aligned against the Arabidopsis reference genomes using Burrows-Wheeler Aligner [69] with default parameters and filtered to keep only reads mapping to the mtDNA or cpDNA. The coverage was extracted with Bedtool genomecov [70] and coverage positions were rounded down to the upper kb. The coverage of each 1 kb-range was normalized relative to 1,000,000 total Arabidopsis reads. For analysis of rearranged sequences based on soft-clipping information, reads pairing to the mtDNA or cpDNA were filtered to only keep those showing a soft-clipping sequence (threshold 20 nucleotides) and no indel (looking for the presence of a S in the CIGAR string without any I,D and

H). The soft-clipping sequences were then extracted with SE-MEI/extractSoftclipped (github. com/dpryan79/SE-MEI) and aligned using bowtie2 [71] against the cpDNA or mtDNA with "—no-mixed" parameters. The positions of the soft-clipping sequences and of their relative reads were rounded down to the upper kb to analyze the location of the rearrangement. Those corresponding to the isomerization that results from the recombination involving the cpDNA large inverted repeats were filtered out. The pipeline is described in detail in https://github. com/ARNTET/Plant_organellar_DNA_recombination.

## Accession numbers

Sequence data from this article can be found in the Arabidopsis Genome Initiative or Gen-Bank/EMBL databases under the following accession numbers: *RADA*, At5g50340; *RECG1*, At2g01440; *RECA2*, At2g19490; *RECA3*, At3g10140; *ANAC044*, At3g01600; *ANAC085*, At5g14490; *KNOLLE*, At1g08560; *CDKB1;2*, At2g38620; *EHD2*, At4g05520; *PLE*, At5g51600; *MYB3R3*, At3g09370; *MYB3R4*, At5g11510; *CYCD3;1*, At4g34160; *CYCD3;2*, At5g67260; *CYCD3;3*, At3g50070; *MCM2*, At1g44900; *MCM3*, At5g46280; *PCNA1*, At1g07370; *PCNA2*, At2g29270; *CDT1a*, At2g31270; *CycB1;1*, At4g37490; *WEE1*, At1g02970; *BRCA1*, At4g21070; *RAD51*, At5g20850; *XRI1*, At5g48720; *SIM*, At5g04470; *SMR1*, At3g10525; *SMR2*, At1g08180; *SMR3*, At5g02420; *SMR4*, At5g02220; *SMR5*, At1g07500; *SMR6*, At5g40460; *SMR7*, At3g27630; *SMR8*, At1g10690; *SMR9*, At1g51355; *SMR10*, At2g28870; *SMR11*, At2g28330; *SMR12*, At2g37610; *SMR13*, At5g59360; AOX1, NM_001249237; RBCS, At1g67090.

## Supporting information

**S1 Fig. Sequence alignment.** Sequence alignment between representative land plant RADA sequences and RadA from proteobacteria and cyanobacteria. The Zinc-finger and KNRFG RadA-specific motif are shaded in yellow and blue respectively, and the Walker A and B motifs in green.
(TIFF)

**S2 Fig. Spatial localization of GFP fusions in leaf cells.** The full length RADA protein or just its N-terminal 174 residues were cloned fused to GFP (RADA:GFP and N-RADA:GFP respectively) and expressed in Arabidopsis under the constitutive ubiquitin promoter. **(A)** Orthogonal view of a rosette leaf (abaxial side) from a plant expressing RADA:GFP, showing that the GFP fluorescence is mainly restricted to surface epidermal cells. The visible green dots correspond to chloroplast nucleoids, which were much brighter than mitochondria. **(B)** Predominant localization of RADA:GFP within vascular tissues as compared to neighboring mesophyll cells. The yellow color of chloroplasts results from the important background of chlorophyl, under the microscope parameters needed to visualize the GFP fluorescence. **(C)** As in (A), but in leaf of plant expressing N-RADA:GFP (just the N-terminal sequence of RADA fused to GFP), showing the same spatial localization.
(TIFF)

**S3 Fig. Tissue-specific expression of Arabidopsis *RADA*.** **(A)** Promoter-GUS fusion results, showing predominant activity of the *RADA* promoter in very young leaves, in sepals, in anthers filament and in the stigmata. The scale bar is 1 mm. **(B)** Expression results extracted from Genevestigator (https://genevestigator.com/).
(TIFF)

**S4 Fig. Expression and characterization of recombinant RADA.** **(A)** Coomassie gel staining analysis of the recombinant proteins RADA protein and Walker mutant K201A, purified by affinity and gel filtration. **(B)** Gel filtration on Superdex S200 showed that RADA purified as

two peaks of high molecular weight. (**C**) Dynamic light scattering of the protein fraction from peak 2 shows that it is monodispersed and corresponding to a size of about 340 kDa, consistent with a hexameric RADA molecule. (**D**) EMSA analysis of the binding to an ssDNA oligonucleotide. Fractions corresponding to both peaks formed complexes of the same size, although fractions of peak 1 formed predominantly higher molecular weight complexes.
(TIFF)

**S5 Fig. Reduced fertility of *radA* plants.** (**A**) Differential interference contrast images of ovules in crosses between *radA-1* flowers and WT pollen. Black arrowheads indicate central cell and egg cell nuclei in unfertilized ovules. White arrowheads indicate developing embryos in fertilized ovules, at the 2–8 cells globular stage, three days after pollination (DAP). Only 16% of *radA* ovules could be fertilized. Scale bar is 50 μm. (**B**) At seven DAP the pollinated pistils did not develop further.
(TIFF)

**S6 Fig. Transmission electron microscope images of *radA* as compared to WT.** Cells from leaves of same size showed morphologically normal chloroplasts (cp) in *radA*, while mitochondria (mt) were enlarged and less electron dense as compared to mitochondria from WT cells.
(TIFF)

**S7 Fig. No significant effect of the *radA* mutation on the stability of the cpDNA.** Leaf DNA from four independent 4-week-old *radA-1* plants and of two WT plants was sequenced on a MySeq Illumina system (2x150 bp paired-end) and reads corresponding to the cpDNA were investigated for possible changes in cpDNA stability. (**A**) Normalized cpDNA coverage, showing comparable copy number of the cpDNA in WT and in *radA*, and no apparent changes in sequences stoichiometry. A schematic representation of the cpDNA is shown above, indicating the large and small single-copy regions (LSC and SSC, respectively) and a single copy of the inverted repeated region (IR). (**B**) Abundance of reads identified as potentially corresponding to rearranged cpDNA molecules, using soft-clipping information. A slight increase, up to twofold as compared to WT, was identified in the *radA* plants. (**C**) Schematic representation of the workflow for identification of cpDNA rearrangements. Blue background boxes text manipulation steps, while yellow and red background boxes represent quality filtering and mapping, respectively. Sequencing was performed on both ends of DNA fragments (R1 and R2). BWA: Burrows-Wheeler Aligner. R1 & R2: Paired-end sequencing read1 and read2. The pipeline is described in https://github.com/ARNTET/Plant_organellar_DNA_recombination.
(TIFF)

**S8 Fig. Hemicomplementation of *radA*.** (**A**) Schematic representation of the constructs used for testing hemicomplementation of the *radA* phenotypes, with a construct only targeting RADA to mitochondria, by fusion to the targeting sequence of alternative oxidase (AOX1), and with a construct targeting RADA to plastids, by fusion to the targeting sequence of the small subunit of Rubisco (RBCS). (**B**) Growth phenotypes of four-week-old hemicomplemented plants in a segregating T2 population, as compared to WT and *radA-1*. Only partial complementation was observed with the AOX1:RADA construct, but plants complemented with RBCS:RADA showed little or no complementation of the mutant phenotype. (**C**) Relative quantification of the accumulation of sequence L-1/2 resulting from recombination across repeats L (as in Fig 6C), in *radA-1* and in hemicomplemented plants. Results are in a log2 scale.
(TIFF)

**S9 Fig. No significant change in cpDNA copy number in *recA3 radA*.** Quantification by qPCR of the relative copy number of the cpDNA genes *ClpP* and *NadH* in *recA3-2 radA-1* double mutants. The results are the mean from three independent plants, ± *SD*. (TIFF)

**S1 Table. List of primers and numerical values underlying graphs shown in Figs 4B, 6B, 6C, 8A, 9C, 9D, 9E, 9F, 9H and S7.** (XLSX)

## Acknowledgments

We are grateful to Dr. Sandra Noir for help with flow cytometry, to Dr. Esther Lechner for vectors and technical advice and to Dr. Patrice Polard and Dr. Bénédicte Michel for helpful discussions.

## Author Contributions

**Conceptualization:** Cécile Raynaud, José M. Gualberto.

**Data curation:** Arnaud Fertet.

**Formal analysis:** Nicolas Chevigny, Cécile Raynaud, José M. Gualberto.

**Funding acquisition:** André Dietrich, José M. Gualberto.

**Investigation:** Nicolas Chevigny, Frédérique Weber-Lotfi, Anaïs Le Blevenec, Cédric Nadiras, Mathieu Erhardt, Cécile Raynaud, José M. Gualberto.

**Methodology:** Nicolas Chevigny, Frédérique Weber-Lotfi, Anaïs Le Blevenec, Marc Bichara, Cécile Raynaud, José M. Gualberto.

**Project administration:** André Dietrich, José M. Gualberto.

**Resources:** Marc Bichara.

**Software:** Arnaud Fertet.

**Supervision:** José M. Gualberto.

**Writing – original draft:** Nicolas Chevigny, Cécile Raynaud, José M. Gualberto.

**Writing – review & editing:** Frédérique Weber-Lotfi, André Dietrich, Cécile Raynaud, José M. Gualberto.

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
