## [Decision Letter · Decision Letter 0]

19 Mar 2022

Dear Dr Gualberto,

Thank you very much for submitting your Research Article entitled 'RADA-dependent branch migration has a predominant role in plant mitochondria and its defect leads to mtDNA instability and cell cycle arrest' to PLOS Genetics.

The manuscript was fully evaluated at the editorial level and by independent peer reviewers. The reviewers appreciated the attention to an important topic but identified some concerns that we ask you address in a revised manuscript

We therefore ask you to modify the manuscript according to the review recommendations. Your revisions should address the specific points made by each reviewer.

[LINK]

Yours sincerely,

Ian Small

Guest Editor

PLOS Genetics

Claudia Köhler

Section Editor: Plant Genetics

PLOS Genetics

Based on the reviewers' comments, I conclude that this manuscript should be acceptable for publication in PLOS Genetics after some minor revisions to the text and figures. No new experimentation is required, just in a couple of cases some re-analysis of the data — notably the use of more sophisticated methods for phylogenetic inference for the data in Figure 1 (Reviewer 1 recommends a couple of suitable software packages for doing this; personally, I would recommend iqtree2 from http://www.iqtree.org). Other than that, all the reviewers' suggestions are for improvements to the clarity of the text or figures and I think all of them should be considered carefully.

Reviewer's Responses to Questions

**Comments to the Authors:**

Reviewer #1: The manuscript focuses on the role of RadA on plant organelles. Numerous experimental assays provide solid evidence for the stated conclusions. I commend the authors for the wide diversity of high quality and sophisticated analyses done resulting in a thorough characterization of RadA in plants.

The main results include:

RADA is targeted to mitochondria and chloroplast

In Arabidopsis, GFP-protein is observed in the epidermis of rosette leaves.

It exhbits similar activity as bacterial RadA, as the plant radA complements the bacterial radA mutant.

Plant RADA preferentially binds ssDNA and accelerates the in vitro strand-exchange reaction.

RADA cannot initiate strand invasion.

It promotes branch migration in vitro, even in the absence of RECA.

The loss of RADA affects plant development and fertility, in contrast to recG1 mutants with no notable phenotype.

Double mutant radA recG1 does not show a synergistic effect.

RADA has a predominant role in mtDNA recombination.

Double mutant radA recA3 is lethal.

In yeast 2 hybrid: RADA interacts with RECA2, but not with RECA3 or RECA1.

RADA mainly participates in the RECA2-recombination pathway of plant organelles.

Loss of RAD promotes mitochondrial ectopic asymmetric recombination across intermediate repeats through the BIR pathway. It leads to the formation of subgenomic molecules that replicate autonomously. The CP has no intermediate repeats and no effect was observed in these mutants.

The severe phenotypes of radA mutants seem to result from a mitochondrial retrograde signal promoting the inhibition of cell cycle progression.

Comments and suggestions for improvement:

Introduction:

In the first paragraph the involvement of different size repeats in recombination is introduced. However, this statement should be rephrased to be more accurate.

What is the evidence for illegitimate recombination of repeats of 100-500 bp in size in wild type plants (which is implied in the introduction)? These repeats could be involved homologous recombination as they have sufficient length of similar sequences, though at lower frequency than the large repeats.

If so, I would clarify that illegitimate recombination would occur at short repeats, instead of intermediate-size repeats.

Also, I understand that HR between large repeats is also considered ectopic. Please, clarify the use of ectopic when referring to IRs.

The first sentence of the second paragraph should be revised. “Defect of the” should probably be deleted.

Results:

Figure 1A. Phaeophytes (now Phaeophyceae) are part of the monophyletic Stramenopiles. See Adl et al. 2019 doi:10.1111/jeu.12691

Fig1A legend: what is the meaning of the numbers on branches? and of the scale bar?

Also Materials and Methods:

Phylogenetic inference should be done using the high-performance methods based on ML and MP criteria, instead of NJ. The size of the alignment should be easily analyzed under ML with 100 to 1,000 bootstrap pseudoreplicates using PhyML or RaxML. Boosstrap support values should be shown on the branches of the best tree.

Figure 2. In the legend, 2D is mentioned, which is not present and 2C is not mentioned. They type of tissue shown should be detailed in the legend.

“We also found RADA:GFP mainly localized in the epidermis and vascular tissue of the Arabidopsis rosette leaves (Supplemental Figure S2A).”

Figure S2 does not show evidence for the localization in vascular tissue. Where is that evidence presented?

Figure S2 legend; clarify if these are rosette leaves. Is vascular tissue shown? Explain N-RADA:FGP versus RADA:GFP.

“However, our observations of RADA:GFP expressed in different plant tissues (leaves, roots

and flowers) did not give any hint that Arabidopsis RADA could be also targeted to the

nucleus.”

So far in the text, there is no evidence of the expression in different plant tissues. If you refer to the information in Fig. S3, indicate so or move to this statement to the following paragraph.

Figure 4:Explain the meaning of the annotations in the x-axis in the legend of figure 4F.

“We tested by qPCR the accumulation of crossover products for repeats F, L and EE and, as expected, in all plants there was a significant increase in crossover products versus WT levels (Figure 6C),”

I disagree with the use of the term “crossover products” as it implies that those alternative arrangements are the result of crossover. Instead, and as stated later, they could and likely are the result of BIR. Thus, I would change crossover products to alternative arrangements or conformations.

“which could be because of alternative mtDNA repair by the error-prone break induced

replication (BIR) pathway [6, 7].”

None of the HR pathways are error free, although BIR is usually regarded as the most error-prone. The types of mutations observed in HR repair include base pair substitutions, indels, and complex chromosome rearrangements. Did you observe any? did you observe polymorphisms in the DNAseq reads of the regions resulting from BIR?

“. Hoever, these values are misleading because compared to the basal levels that exist in WT plants.”

Please fix the grammar to improve this sentence.

Bioinformatic analysis:

Please include the parameters used for the alignment using BWA. Note that depending on the parameters, a read that aligns to two regions of the reference sequence (i.e. a repeat) could be aligned twice (to both regions) or only once (i.e. randomly to only one of the two copies of the repeat). thus, the resulting figure and interpretation would be quite different. Also, a range of mismatches may be allowed for alignment depending on the parameters provided. Thus, the parameters provide key information to understand the results.

Same for bowtie alignment.

It is not clear what do you mean by “short-clipping sequence”? Is it part of the Illumina read that does not align with the reference sequence? Which % of the read can it represent?

Why is this section focused on the analysis of the cpDNA and not the mtDNA? I’m not sure if or why the mtDNA was analyzed differently.

“Rather, for most transcripts an increased accumulation was observed, as compared to WT plants of the same size (Figure 8A), up to 8-fold in the case of the rps4 transcript.’

An statistical test would be useful to detect significant differences with the wild-type. Also, large increases in expression levels are “observed effects” of radA-induced mtDNA instability. Thus, I disagree with the title of this section.

Reviewer #2: This is a very interesting manuscript that provides insights into some of the long-standing questions regarding proteins and mechanisms involved in plant mitochondrial DNA recombination and genome maintenance. The distinct role of RadA is shown in this work, and the combination of methods and approaches used lead to data that support the conclusions of the work. I found the manuscript to flow well and the work seems quite complete to me. I do not have any concerns about the experiments or data, and feel that the conclusions made are supported by the work.

The manuscript could be improved for clarity by careful proofreading and editing. Two examples:

1. Second paragraph starts 'Defect of the factors...' I think 'defect' is not the correct word to use here and is confusing. Please reword.

2. p. 9 last line, typo, should be 'identified' not 'identifies'

Reviewer #3: The authors of this manuscript have characterized an organellar RadA homolog in the model plant Arabidopsis. Using a nice combination of in vitro biochemistry and in vivo studies, they confirm that the gene encodes a true ortholog of the bacterial RadA. While the main function in branch migration has been retained, the work also uncovers some new features that seem to differ from bacterial RadA proteins (e.g., independence of interaction with RecA). The manuscript contains a large amount of experimental data, most of which are of high quality.

I have only a few comments for the authors to consider:

1. The authors suspect the punctate protein localization in chloroplasts to correspond to nucleoids, and used fluorescent protein-tagged PEND to substantiate this conclusion. Whether or not (overexpressed!) PEND::dsRED is a reliable nucleoid marker is somewhat questionable. It would have been more convincing had the authors conducted counter-staining with DAPI, which is generally accepted as a more reliable method of identifying chloroplast nucleoids.

2. For the hemicomplementation experiments, the authors used the RBCS TP to target the protein exclusively to chloroplasts. However, the RBCS TP has been shown to exhibit some leakiness in that it causes (low-level) mistargeting to mitochondria. See, e.g.: Tabatabaei I, et al. (2019) A highly efficient sulfadiazine selection system for the generation of transgenic plants and algae. Plant Biotechnol J 17: 638-649. This problem should be acknowledged and discussed.

Minor points:

- Introduction, 2nd paragraph: The first sentence (‘Defect of the factors…’) seems to require some rewording.

- Introduction, last sentence: I don’t think that retrograde signaling ‘mobilizes genes’. I suspect the authors mean ‘activates genes’?

- p. 9: The statement on enlarged mitochondria and the connection of this ultrastructural phenotype to mitochondrial dysfunction and/or impaired gene expression should be referenced and discussed in the context of the existing literature on mitochondrial mutants displaying similar phenotypes (e.g., Zhou W, et al. (2015) Multiple RNA processing defects and impaired chloroplast function in plants deficient in the organellar protein-only RNase P enzyme. PLoS One 10: e0120533).

- The manuscript should be carefully proofread for spelling and grammar (e.g., p.10 ‘Hoever, these values…)’; p.11 ‘…maintenance ant that the loss…’, etc.).

**Have all data underlying the figures and results presented in the manuscript been provided?**

Reviewer #1: Yes

Reviewer #2: Yes

Reviewer #3: Yes

PLOS authors have the option to publish the peer review history of their article (what does this mean?). If published, this will include your full peer review and any attached files.

Reviewer #1: No

Reviewer #2: No

Reviewer #3: No

---

## [Editor Report · Decision Letter 1]

14 Apr 2022

Dear Dr Gualberto,

We are pleased to inform you that your manuscript entitled "RADA-dependent branch migration has a predominant role in plant mitochondria and its defect leads to mtDNA instability and cell cycle arrest" has been editorially accepted for publication in PLOS Genetics. Congratulations!

Yours sincerely,

Ian Small

Guest Editor

PLOS Genetics

Claudia Köhler

Section Editor: Plant Genetics

PLOS Genetics

Comments from the reviewers (if applicable):

Thank you for the comprehensive and detailed responses to all the reviewers' comments. I believe that all the necessary improvements have been made to the manuscript and that it is now acceptable for publication.

**Data Deposition**

http://datadryad.org/submit?journalID=pgenetics&manu=PGENETICS-D-22-00131R1

**Press Queries**

---

## [Editor Report · Acceptance letter]

6 May 2022

PGENETICS-D-22-00131R1 

RADA-dependent branch migration has a predominant role in plant mitochondria and its defect leads to mtDNA instability and cell cycle arrest 

Dear Dr Gualberto, 

We are pleased to inform you that your manuscript entitled "RADA-dependent branch migration has a predominant role in plant mitochondria and its defect leads to mtDNA instability and cell cycle arrest" has been formally accepted for publication in PLOS Genetics! Your manuscript is now with our production department and you will be notified of the publication date in due course.

With kind regards,

Anita Estes

PLOS Genetics

On behalf of:
